

# Substantial root-zone water storage capacity observed by GRACE and GRACE/FO

Meng Zhao[1], Erica L. McCormick[2], Geruo A[3], Alexandra G. Konings[2], Bailing Li[4,5]

[1]Department of Earth and Spatial Sciences, University of Idaho, Moscow, ID 83843, U.S.
[2]Department of Earth System Science, Stanford University, Palo Alto, CA 94305, U.S.
[3]Department of Earth System Science, University of California, Irvine, CA 92617, U.S.
[4]NASA Goddard Space Flight Center, Greenbelt, MD 20771, U.S.
[5]Earth System Science Interdisciplinary Center, University of Maryland, College Park, MD 20742, U.S.

*Correspondence to*: Meng Zhao (mengz@uidaho.edu)

**Abstract.** Root-zone water storage capacity ($S_r$) - the maximum water volume that can be held in the plant root zone - bolsters ecosystem resilience to droughts and heat waves, influences land-atmosphere exchange, and controls runoff and groundwater recharge. However, $S_r$ is difficult to measure, especially at large spatial scales, hindering accurate simulations of many biophysical processes, such as photosynthesis, evapotranspiration, tree mortality, and wildfire risk. Here, we present a global estimate of $S_r$ using direct measurements of total water storage (TWS) anomalies from the Gravity Recovery and Climate Experiment (GRACE) and GRACE Follow-On satellite missions. We find that the median $S_r$ value for global vegetated regions is at least 220 ± 40 mm, which is over 50% larger than the latest estimate derived from tracking storage change via water fluxes, and 380% larger than that calculated using the soil and rooting depth parameterization. Parameterizing a global hydrological model with our $S_r$ estimate improves TWS and evapotranspiration simulations across much of the globe. Furthermore, our $S_r$ estimate, based solely on hydrological data, correlates realistically with an independent vegetation productivity dataset, underscoring the robustness of our approach. Our study highlights the importance of continued refinement and validation of $S_r$ estimates and provides a new pathway for further exploring the impacts of $S_r$ on water resource management and ecosystem sustainability.

## 1 Introduction

During periods of insufficient precipitation, vegetation relies on water stored underground to survive (Miguez-Macho and Fan, 2021). The larger the root-zone water storage capacity ($S_r$), the more water plants can store during wet periods for use in droughts (Teuling et al., 2006). $S_r$, therefore, plays an important role in regulating ecosystem resilience to droughts and heat waves and affecting wildfire outbreaks and mortality risk (Callahan et al., 2022; Chen et al., 2013; Goulden and Bales, 2019; Hahm et al., 2019; Humphrey et al., 2018; Stocker et al., 2023). It is also an essential parameter for modeling plant
carbon uptake, transpiration, soil evaporation, streamflow, and groundwater (Maxwell and Condon, 2016; Zhao et al., 2022;



Peterson et al., 2021). Despite its critical role in modulating the carbon and water cycles, global patterns of $S_r$ remain poorly characterized.

The $S_r$ is typically calculated as the integration of plant rooting depth and soil texture-dependent water-holding capacity (Seneviratne et al., 2010; Vereecken et al., 2022; Speich et al., 2018; Federer et al., 2003). However, this approach

(hereafter referred to as the rooting depth-based estimation) suffers from uncertainties associated with plant rooting depth and substrate hydraulic properties, particularly at depth, undermining the accuracy of the calculated $S_r$ (Vereecken et al., 2022; Novick et al., 2022). Additionally, it overlooks a significant contribution to $S_r$ from plant roots extracting moisture stored in weathered bedrock in the form of rock moisture (Rempe and Dietrich, 2018; Mccormick et al., 2021) and groundwater (Maxwell and Condon, 2016; Fan et al., 2017).

More recently, Earth observations of precipitation (P) and evapotranspiration (ET) have been used to estimate $S_r$. Several studies (Stocker et al., 2023; Wang-Erlandsson et al., 2016; Gao et al., 2014; Mccormick et al., 2021) have proxied Sr using the maximum cumulative difference in ET and P during dry periods (when ET > P), which reflects the largest water volume that an ecosystem has withdrawn from its root zone. This method (hereafter referred to as the water deficit-based estimation) is based on mass balance and thus eliminates the need for information about plant access to rock moisture and

groundwater, rooting depth, and soil and bedrock hydraulics. However, obtaining accurate P and ET data is challenging at scale (Sun et al., 2018; Miralles et al., 2016), and errors in these data can accumulate and deteriorate $S_r$ calculations. Here, to avoid this shortcoming, we estimated root-zone storage dynamics directly from total water storage (TWS) anomalies measured by the Gravity Recovery and Climate Experiment (GRACE) and GRACE Follow-On (GRACE-FO) satellite missions (hereafter GRACE/FO). With these direct observations, we characterized global patterns of $S_r$ and found that both the rooting

depth-based estimate and the water deficit-based estimate have significantly underestimated $S_r$.

## 2 Materials and methods

### 2.1 GRACE/FO TWS

We use monthly measurements of the TWS anomaly from GRACE for the years 2002-2017 and from GRACE-FO for the years 2018-2022. These measurements were obtained from the Jet Propulsion Laboratory (JPL) RL06 solutions

(Watkins et al., 2015; Wiese et al., 2016), which provide monthly average anomalies of the gravity field over an equal-area $3° \times 3°$ mass concentration block (mascon). We opted for the JPL mascon solutions because each JPL mascon is relatively uncorrelated with neighboring mascons and thus offers more localized spatial variations than other mascon solutions and the spherical harmonic solutions (Watkins et al., 2015; Wiese et al., 2016). We did not fill the 11-month gap (July 2017 to May 2018) between GRACE and GRACE-FO. However, we linearly interpolated other missing months from the nearest previous

and subsequent non-missing values (Rodell et al., 2018; Zhao et al., 2021). Because we aimed to estimate root-zone storage capacity $S_r$, we only included mascon locations with over 50% fractional vegetation cover based on the land cover product





(MCD12Q1) version 6.1 from the Moderate Resolution Imaging Spectroradiometer (MODIS) (Sulla-Menashe and Friedl, 2018).

## 2.2 $S_r$ from TWS drawdown and uncertainty estimate

Ecosystem use of land water storage for ET is represented in the TWS drawdown, that is, a consecutive decline in TWS anomaly despite seasonal or intermittent recharge. An example is illustrated in Fig. 1 at a mascon location in southern Idaho, where the largest TWS drawdowns are annotated. From the water balance, a TWS drawdown over a time-period Δt is equal to:

$$\Delta TWS = P - ET - R \qquad (1)$$

where P, ET, and R are the total precipitation, total evapotranspiration, and net runoff out of the mascon over Δt, respectively.

Based on eq (1), when precipitation exceeds runoff (P - R > 0), any TWS drawdown (or negative ΔTWS) must be influenced by a change in storage due to ET. To determine if precipitation exceeds runoff during GRACE/FO-observed TWS drawdowns, we compared R estimates from a multi-forcing observation-based global runoff reanalysis (Ghiggi et al., 2021) to P estimates from the Global Precipitation Climatology Project (Gebremichael et al., 2003). We found that in nearly all analyzed mascon locations, the cumulative sum of P - R is positive during at least the five largest TWS drawdowns (Fig. A1), confirming these

TWS drawdowns reflect root-zone water storage consumed by ecosystems.

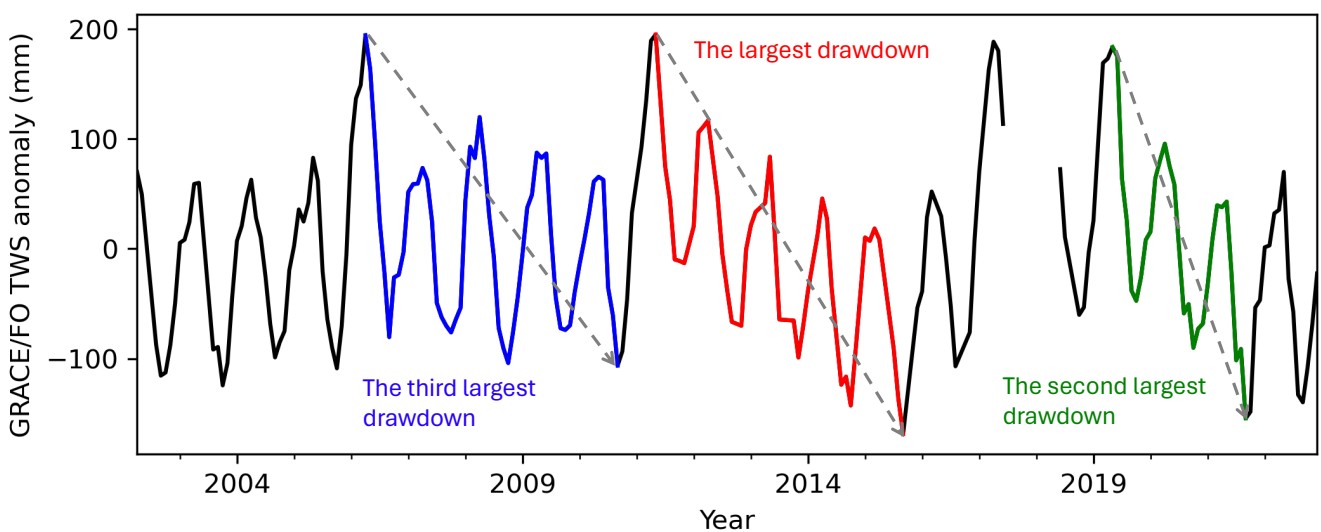

**Figure 1**. Example of the three largest TWS drawdowns at a mascon location in southern Idaho.

We estimated root-zone water storage capacity $S_r$ to be the largest TWS drawdown during the record period of

GRACE/FO (denoted as SrGRACE/FO). To avoid overestimating $S_r$, we removed the impact of groundwater pumping, snow, and surface water on TWS drawdowns. Groundwater pumping, often manifested as a negative long-term trend in the TWS



time series (Rodell et al., 2018; Rodell et al., 2009; Feng et al., 2013), is a human-made withdrawal of water resources. To avoid conflating this drawdown with $S_r$, we first calculated the TWS trend by simultaneously fitting an annual and a semiannual signal, a linear trend, and a constant to the GRACE/FO time series (Fig. A2). Then, we assumed any negative trend was 85 attributable to groundwater pumping and removed the negative trend from the original GRACE/FO time series before calculating the TWS drawdowns. In high-latitude and mountainous regions, the maximum TWS anomaly during drawdowns may include snow. To avoid attributing snow storage to root-zone water storage, we first determined the largest drawdown from the full GRACE/FO time series and then calculated $S_r$ using the maximum and minimum TWS anomaly with a monthly mean air temperature above 5°C. We obtained air temperature data from the fifth-generation European Centre for Medium-90 Range Weather Forecasts atmospheric reanalysis of the global climate (ERA5) (Hersbach et al., 2020). Following Wang et al. (2023), we used total runoff from Ghiggi et al. (2021) as a proxy for surface water storage change and removed it from TWS drawdowns to isolate the subsurface contributions to the GRACE/FO signal. Note that total runoff from Ghiggi et al. (2021) stopped in 2019, and we used monthly climatology values between 2002 and 2019 to extend the data to 2022 and align with the GRACE/FO record length. Other contributions to TWS drawdowns, such as changes in water intercepted by leaf and 95 branch surfaces and internal plant water storage, are too small to be detected by GRACE/FO (Rodell et al., 2005).

We calculated the random error of $S_r^{GRACE/FO}$ by adding errors of the two GRACE/FO measurements and the uncertainty of groundwater pumping and surface water signals in quadrature. To calculate the GRACE/FO measurement error, we used the formal error product provided by the JPL mascon solutions (Watkins et al., 2015; Wiese et al., 2016). For the uncertainty of groundwater pumping and surface water signals, we assumed a ±50% error on the magnitude of our calculated 100 signals following Zhao et al. (2021). This assumption implies that the uncertainty range is equal to the signals themselves, leading to a likely conservative error estimate.

### 2.3 Comparison to other $S_r$ estimates

We compared our $S_r^{GRACE/FO}$ estimate to two other $S_r$ datasets. These datasets represent the typical rooting depth × soil texture-dependent water holding capacity approach (referred to as $S_r^{RD \times WHC}$) and the water deficit accumulation approach 105 (referred to as $S_r^{accum}$). We chose the $S_r^{accum}$ estimate from Stocker et al. (2023) because it used the latest Earth observation-constrained estimates of precipitation and evapotranspiration. We used their "$S_{CWDX80}$" product which was estimated based on cumulative water deficit extremes occurring with a return period of 80 years. We calculated $S_r^{RD \times WHC}$ using existing datasets on rooting depths and soil texture. The $RD \times WHC$ approach requires effective rooting depths (Federer et al., 2003; Speich et al., 2018; Stocker et al., 2023; Bachofen et al., 2024). We obtained effective rooting depths from Yang et al. (2016), who 110 retrieved them using an analytical model that balances the marginal carbon cost and benefits of deeper roots. Soil water holding capacity is calculated based on soil texture information from the Harmonized World Soil Database version 1.2 (Wieder et al., 2014) and pedo-transfer functions based on Balland et al. (2008). The Harmonized World Soil Database provides information for depths of 0-0.3 m and 0.3-1 m. For depths greater than 1 m, we assume texture values from the 0.3-1 m depth following



Stocker et al. (2023). For consistency, we spatially averaged both $S_r^{accum}$ and $S_r^{RD \times WHC}$ estimates to match the GRACE/FO
spatial scale ($3° \times 3°$).

**2.4 Evaluation using the USGS monthly hydrologic model**

To evaluate the relative accuracy of $S_r^{GRACE/FO}$, $S_r^{accum}$ and $S_r^{RD \times WHC}$, we used each of them to separately parameterize a hydrologic model, labeled as $HydroModel(S_r^{GRACE/FO})$, $HydroModel(S_r^{accum})$, and $HydroModel(S_r^{RD \times WHC})$, respectively. Then, we compared the performance of the three models, assessed by their accuracy in simulating observations of TWS and ET. The
atmospheric forcing data and model parameters used in all simulations were identical except for $S_r$. Therefore, their relative model performance demonstrates the differential accuracy between the three estimates. A monthly hydrologic model developed by the United States Geological Survey (USGS) (Mccabe and Markstrom, 2007) was used due to its simplicity and transparency about physical processes. Specifically, the model relies on a straightforward specification of $S_r$ as a "water bucket" depth rather than indirectly through prescribed rooting depth, soil texture, and pedo-transfer functions across the profile. This
allows us to parameterize the model directly with $S_r^{GRACE/FO}$, $S_r^{accum}$, and $S_r^{RD \times WHC}$. The USGS model was run at each GRACE mascon location with air temperature forcing from ERA5 and precipitation forcing from GPCP. We used climate forcing from 1993 to 2001 to spin up the model and performed water cycle simulations for the study period from 2002 to 2022. No calibrations were carried out.

We compared the performance between $HydroModel(S_r^{GRACE/FO})$, $HydroModel(S_r^{accum})$, and $HydroModel(S_r^{RD \times WHC})$
in capturing observed anomalies in TWS and ET. We opted for TWS anomalies as a comparison because they are directly observable (by GRACE/FO) and are most relevant to the root-zone storage process. As the USGS model does not provide a standard output variable for TWS, we used the sum of total root-zone water storage and surface snow amount as an approximation of it, following previous studies (Jensen et al., 2019; Scanlon et al., 2018). Due to a lack of groundwater compartment, the USGS model may underestimate large decadal declining and rising water storage trends relative to
GRACE/FO (Scanlon et al., 2018). To minimize this impact on our model comparison, we detrended both the GRACE/FO TWS time series and the model simulations of TWS. For consistency with GRACE/FO, modeled TWS anomalies were calculated by subtracting the time mean between 2002 and 2022 from the modeled TWS time series. Despite being the same dataset used in calculating $S_r^{GRACE/FO}$, using GRACE/FO as reference data is not circular because we calculated $S_r^{GRACE/FO}$ by taking the difference of only two measurements (i.e., the maximum and minimum TWS values during the largest TWS
drawdown). The complete GRACE/FO time series remains a useful dataset for evaluating model performance.

In addition, we compared model performance in simulating ET anomaly. We noted that existing gridded ET products generally have assumed ecosystem responses to water stress in their algorithms and are thus highly uncertain (Miralles et al., 2016). Most of these algorithms use so-called β-based formulations to model the impact of water stress on transpiration, reducing ET by a multiplicative stress factor β that depends on soil moisture (Trugman et al., 2018). These formulations contain
errors and can have unknown impacts on the model performance evaluation (Tang et al., 2024; Miralles et al., 2016; Pascolini-



Campbell et al., 2020). Instead, we used ET estimates derived from a water balance approach provided by Xiong et al. (2023). They calculated ET using eq (1) for major river basins by generating 4669 probabilistic unique combinations of 23 precipitation, 29 total runoff, and 7 water storage change datasets. These ET estimates are based on mass conservation and thus do not have assumed plant-water relations. We only considered basins with an area extent larger than the nominal

resolution of GRACE/FO (~100,000 km$^2$). As the USGS hydrologic model was run at the mascon scale, we followed Zhao et al. (2022) to aggregate basin-scale modeled ET from mascon scale model outputs. We first identified all mascons that fully or partially cover a given basin and calculated the percentage of the total basin area covered by each mascon. We then used these percentage values as weights to calculate the basin-average ET from each mascon model output. Due to biases in existing precipitation and runoff datasets, the water balance-based ET estimates are also biased (Xiong et al., 2023; Rodell et al., 2004;

Swenson and Wahr, 2006; Velicogna et al., 2012). These biases are challenging to correct, as unbiased global ET products are rare and almost non-existent (Miralles et al., 2016; Tang et al., 2024). To reduce its impact on our model evaluation, we focused on ET anomalies and calculated them by removing the corresponding temporal mean from both model output and water balance-based estimates following previous studies (Pascolini-Campbell et al., 2020; Velicogna et al., 2012).

The Nash-Sutcliffe model efficiency coefficient (NSE) was used to assess the predictive skill of each USGS

hydrologic model, which is defined as:

$$\text{NSE} = 1 - \frac{\sum_{t=1}^{T}(X_o^t - X_m^t)^2}{\sum_{t=1}^{T}(X_o^t - \overline{X_o})^2} \qquad (2)$$

where $X$ represents TWS anomaly or ET anomaly, $\overline{X_o}$ is the mean of observed $X$, and $X_o^t$ and $X_m^t$ are observed and modeled $X$ at time $t$, respectively (Nash and Sutcliffe, 1970). An NSE value closer to 1 indicates a better model performance in simulating $X$. An NSE value less than 0 indicates that the mean observed value is a better predictor than the simulated value, suggesting an unsatisfactory model performance (Nash and Sutcliffe, 1970). If *HydroModel*($S_r^{GRACE/FO}$), *HydroModel*($S_r^{accum}$),

and *HydroModel*($S_r^{RD \times WHC}$) all yield negative NSE values, the efficacy of using the USGS hydrologic model to evaluate the relative accuracy of the three $S_r$ estimates is compromised. Here, we focused on mascons and basins where at least one of the three models achieved a positive NSE value.

### 2.5 $S_r$ linkage to vegetation growth

The $S_r^{GRACE/FO}$ is derived from the water balance, but its ecological relevance remains undetermined. To investigate

whether $S_r^{GRACE/FO}$ reflects vegetation water use for growth, we compared it with an independent measure of ecosystem productivity. We used maximum gross primary productivity (*GPP$_{max}$*) to represent the potential GPP when the root zone is saturated with water. We obtained GPP data from the global MODIS and FLUXNET-derived daily GPP product from 2000 to 2020 (Joiner and Yoshida, 2021). We chose this GPP product because it maximized the use of MODIS reflectance bands and demonstrated excellent validation results and agreement with other commonly used GPP products (Joiner and Yoshida, 2020).





**3 Materials and methods**

**3.1 $S_r$ from GRACE/FO ($S_r^{GRACE/FO}$)**

We find a substantial root-zone water storage capacity worldwide. Across the global vegetated domain, $S_r^{GRACE/FO}$ (or the largest TWS drawdown) spans from 22 to 2131 mm (Fig. 2a). The distribution of $S_r^{GRACE/FO}$ is positively skewed, with a median value of 221 mm (129 - 389 mm interquartile range; note that values in parentheses hereafter always refer to the

interquartile range). Larger $S_r^{GRACE/FO}$ is associated with densely vegetated regions like the tropical rainforests, the Southeastern U.S., the Pacific Northwest, and the southern part of China while smaller $S_r^{GRACE/FO}$ is found in sparsely vegetated regions like Central Asia, much of Australia, and some Arctic regions (Fig. 2a). Fig. 2b shows the duration of the maximum TWS drawdown with a global median of 2.8 years (1.6 - 5.2 years). We find no correlation between the duration and the magnitude of the largest TWS drawdown across different regions (Figs. 2a-b). The impact of random error sources on our $S_r^{GRACE/FO}$

estimate remains moderate, with a global median relative error of 18% (13% - 26%) (Fig. 2c).



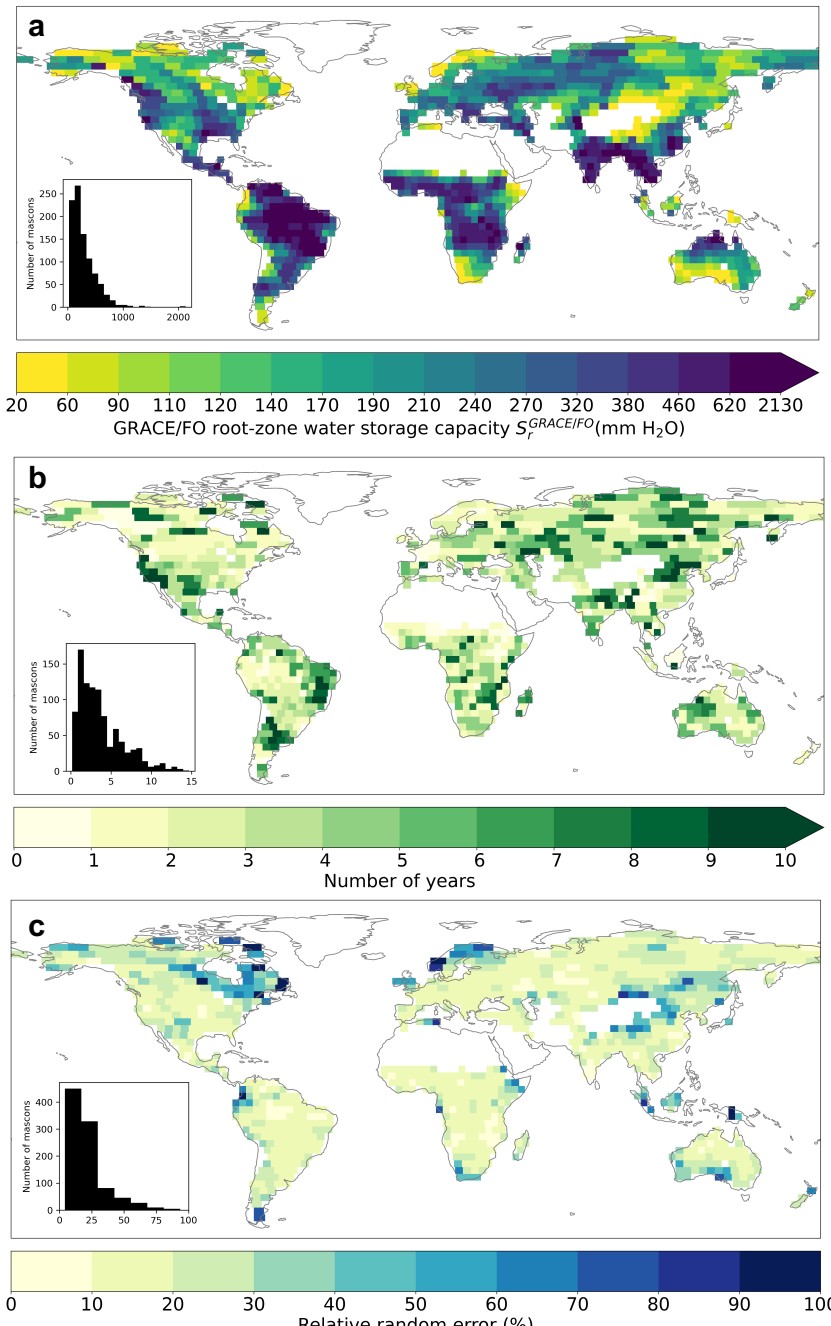

**Figure 2**. $S_r$ estimated from GRACE/FO total water storage (TWS) anomaly. (a) Global patterns of $S_r^{GRACE/FO}$ for Earth's vegetated regions. (b) The duration of the maximum TWS drawdown. (c) Global patterns of the random error of $S_r^{GRACE/FO}$. Insets in (a) - (c) show the histograms of corresponding mapping variables across our study area. White spaces on land represent mascon locations with less than 50% vegetation cover.



To characterize the utilization of root-zone water storage capacity, we compared the second and third-largest TWS drawdowns to $S_r^{GRACE/FO}$. We find that, on average, the second-largest TWS drawdown consumes 83% (71% - 92%) of the $S_r^{GRACE/FO}$ estimate (Fig. 3a), while the third-largest uses 68% (54% - 82%) (Fig. 3b). The average duration of the second- and third-largest TWS drawdowns decreases from 1.6 years (1.1 - 3.2 years) to 1.2 years (0.5 - 1.7 years) (Figs. 3c-d). In about 40% of our analyzed mascons, the longest TWS drawdown period does not coincide with the largest drawdown magnitude. These findings underscore the nuanced dynamics of water storage use within the root zone, suggesting variability in both magnitude and duration across different regions.

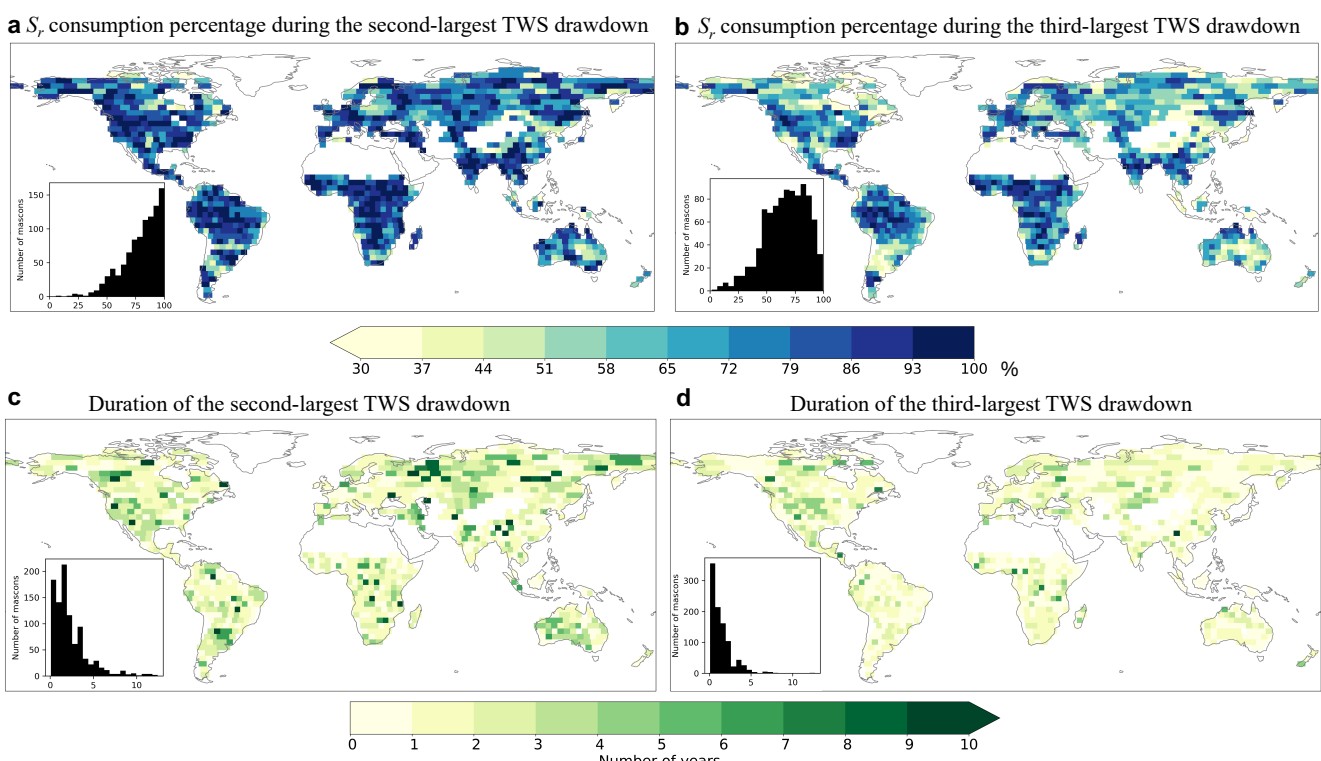

**Figure 3.** Utilization of root zone water storage capacity. (a) and (b) are the $S_r^{GRACE/FO}$ consumption percentages during the second and third-largest TWS drawdowns. (c) and (d) are the duration of the second and third-largest TWS drawdowns. Insets in (a) - (d) show the histograms of corresponding mapped variables.

**3.2 Comparison with other $S_r$ estimates**

Our $S_r^{GRACE/FO}$ estimate is larger than $S_r^{RD \times WHC}$ and $S_r^{accum}$ over much of the globe. Figs. 4a-b show $S_r^{GRACE/FO}$ difference with $S_r^{RD \times WHC}$ and $S_r^{accum}$, respectively. Across the global vegetated domain, $S_r^{GRACE/FO}$ surpasses $S_r^{RD \times WHC}$ in over 90% of mascon locations, with a median value 175 mm (or 380%) higher than that of $S_r^{RD \times WHC}$. The $S_r^{GRACE/FO}$ exceeds $S_r^{accum}$ over 70% of the study area, with a median value 77 mm (or 53% ) higher than that of $S_r^{accum}$, despite exhibiting lower values in drier



climates and lower-biomass regions (Fig. 4b). Notably, these differences are greater than the random error of $S_r^{GRACE/FO}$, emphasizing that the underestimations by $S_r^{RD \times WHC}$ and $S_r^{accum}$ are significant.

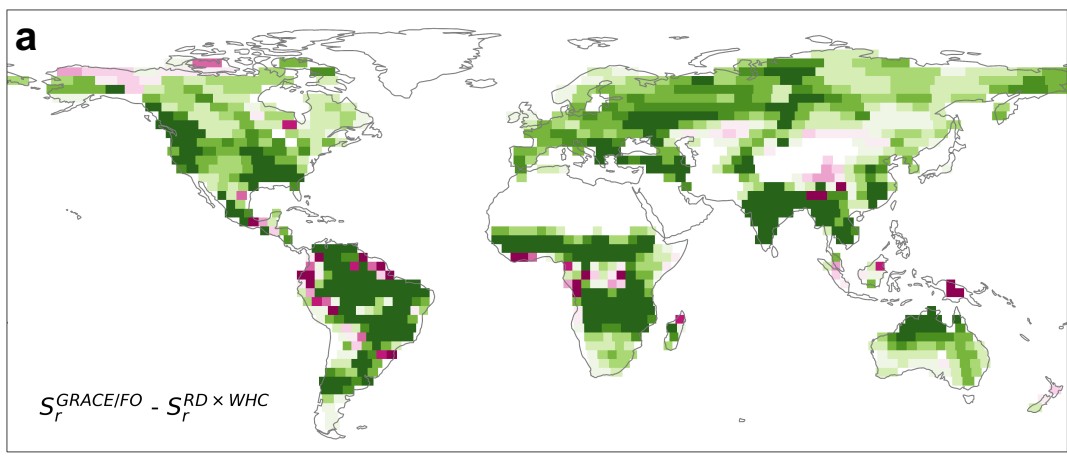

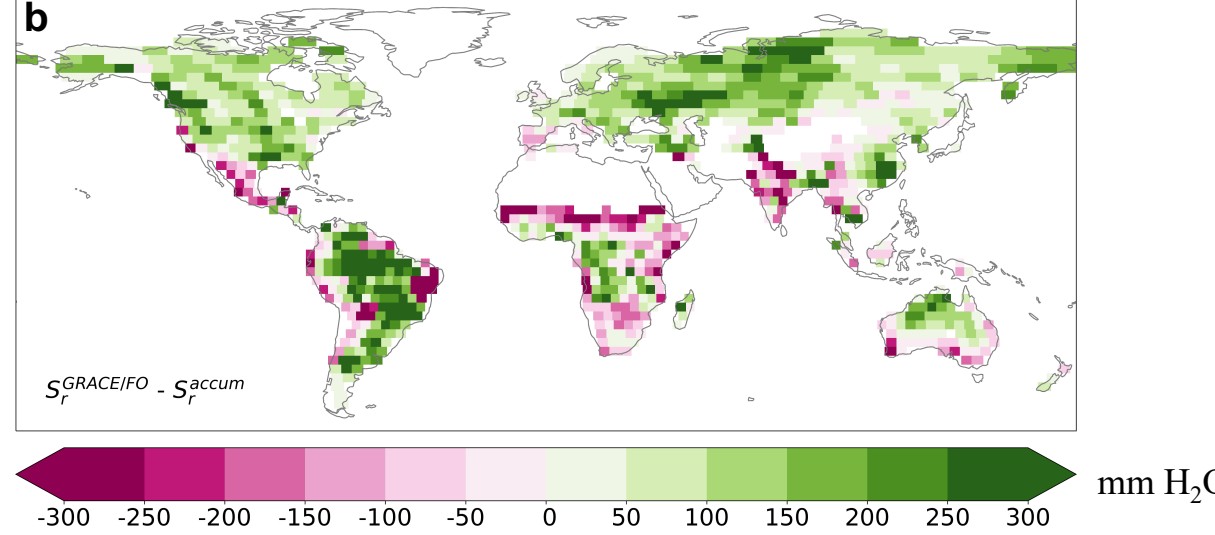

**Figure 4.** $S_r^{GRACE/FO}$ comparison with other datasets. (a) The difference between $S_r^{GRACE/FO}$ and $S_r^{RD \times WHC}$. (b) The difference between $S_r^{GRACE/FO}$ and $S_r^{accum}$.

### 3.3 Implementation in the USGS hydrologic model

To assess whether $S_r^{GRACE/FO}$ is an improvement over $S_r^{accum}$ and $S_r^{RD \times WHC}$, we used each of them to separately parameterize the USGS hydrologic model. We first evaluated the accuracy of $HydroModel(S_r^{GRACE/FO})$, $HydroModel(S_r^{RD \times WHC})$, and $HydroModel(S_r^{accum})$ in replicating the time series of GRACE/FO TWS anomalies. No model attains positive NSE values for approximately 40% of the global vegetated domain (Fig. A3), suggesting the USGS model may not effectively discern the relative accuracy of the three $S_r$ estimates at these locations. However, for the remaining 60%, at least one model achieved a



positive NSE value. In these regions, the average NSE for $HydroModel(S_r^{GRACE/FO})$ is 0.39 (0.23 - 0.59), for
$HydroModel(S_r^{RD×WHC})$ it is -9.33 (-26.66 - 0.30), for $HydroModel(S_r^{accum})$ it is 0.22 (0.09 - 0.56). The $HydroModel(S_r^{GRACE/FO})$
outperformed $HydroModel(S_r^{RD×WHC})$ in terms of NSE values across 89% of these regions and outperformed
$HydroModel(S_r^{accum})$ across 67% of these regions (Fig. 5). For example, at a wet mascon location in the Pacific Northwest (Fig.
6a), the NSE values for $HydroModel(S_r^{GRACE/FO})$, $HydroModel(S_r^{RD×WHC})$, and $HydroModel(S_r^{accum})$ are 0.68, -3.69, and 0.42,
respectively (Fig. 6b). For a dry mascon in Mexico (Fig. 6a), the NSE values for $HydroModel(S_r^{GRACE/FO})$,
$HydroModel(S_r^{RD×WHC})$, and $HydroModel(S_r^{accum})$ are 0.64, -45.6, and 0.54, respectively (Fig. 6c). These results suggest an
improved performance in simulating TWS temporal dynamics when parameterizing root-zone water storage capacity using
$S_r^{GRACE/FO}$ in the hydrologic model. Nevertheless, $HydroModel(S_r^{accum})$ demonstrates superior performance in some drier
climates and lower-biomass regions. For instance, at a mascon in the Horn of Africa (Fig. 6a), the NSE value of
$HydroModel(S_r^{accum})$ is 0.46, significantly higher than that of $HydroModel(S_r^{GRACE/FO})$ and $HydroModel(S_r^{RD×WHC})$, which are -
1.4 and -2.1, respectively (Fig. 6d). The comparison between Fig. 4b and Fig. 5b reveals that the underperformance of
$HydroModel(S_r^{GRACE/FO})$ compared to $HydroModel(S_r^{accum})$ is associated with $S_r^{GRACE/FO}$ consistently being lower than $S_r^{accum}$ in
these arid regions.

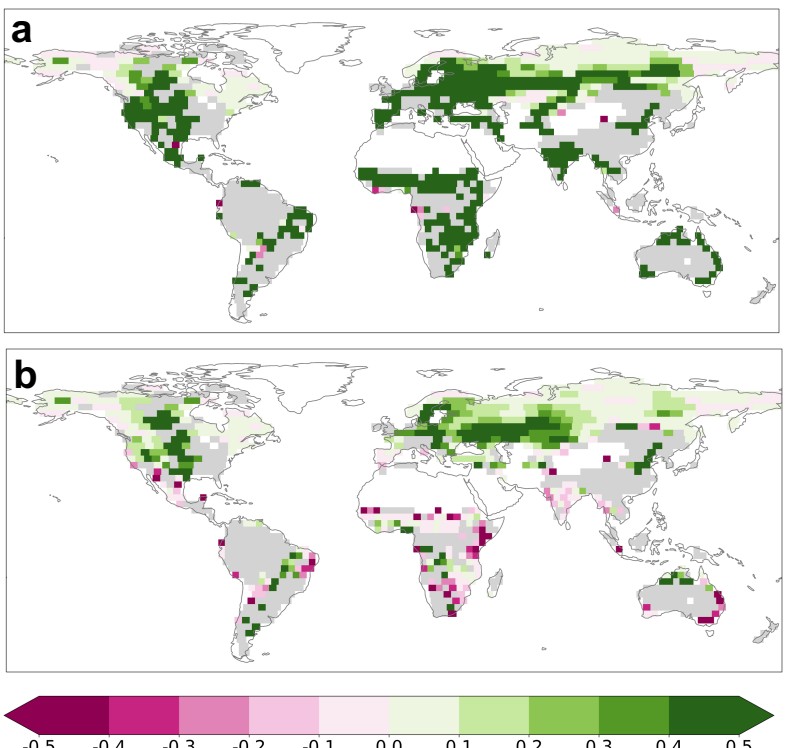

**Figure 5.** Predictive skill differences for TWS anomalies. (a) The NSE difference between $HydroModel(S_r^{GRACE/FO})$ and
$HydroModel(S_r^{RD×WHC})$. (b) The NSE difference between $HydroModel(S_r^{GRACE/FO})$ and $HydroModel(S_r^{accum})$. The gray colors
indicate areas where all models fail to achieve a positive NSE value.





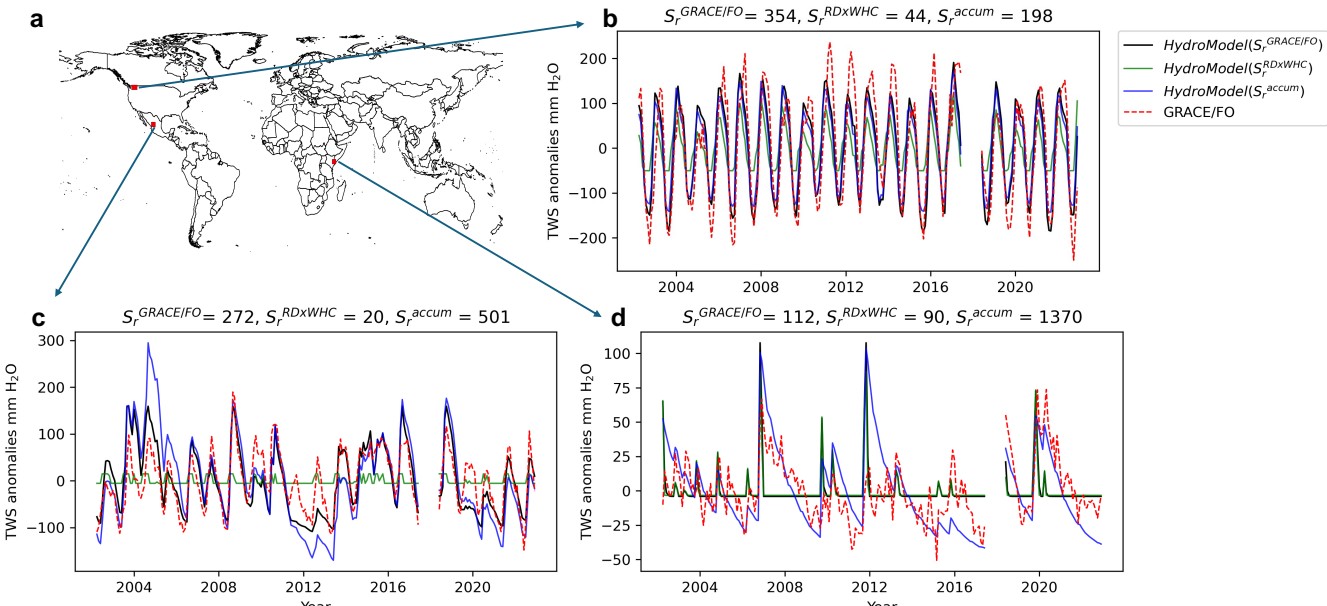

**Figure 6.** Time series comparison between GRACE/FO TWS and model simulations. (a) Location map of the three exemplary
mascons in the Pacific Northwest (b), Mexico (c), and the Horn of Africa (d). The values of $S_r^{GRACE/FO}$, $S_r^{RD\times WHC}$, and $S_r^{accum}$
are annotated on top of (b) - (d).

In addition, we evaluated each model's accuracy in simulating the time series of ET anomalies. The results show that
at least one model achieves a positive NSE value in 48 large river basins (Fig. 7). In these basins, the average NSE for
*HydroModel*($S_r^{GRACE/FO}$) is 0.35 (0.13 - 0.63), for *HydroModel*($S_r^{RD\times WHC}$) it is 0.30 (0.10 - 0.54), and for *HydroModel*($S_r^{accum}$)
it is 0.29 (0.06 - 0.58). Specifically, *HydroModel*($S_r^{GRACE/FO}$) outperformed *HydroModel*($S_r^{RD\times WHC}$) in terms of NSE values
across 37 basins and outperformed *HydroModel*($S_r^{accum}$) across 45 basins (Fig. 7).

Taken together, despite an absence of direct root-zone storage measurements at scale, $S_r^{GRACE/FO}$ notably improves
upon the water deficit-based estimate and the rooting depth-based estimate and reveals a substantially larger root-zone storage
capacity across much of the globe. The improved simulation accuracy of TWS and ET anomalies using $S_r^{GRACE/FO}$ demonstrates
the importance of accurate $S_r$ estimates for hydrological modeling.





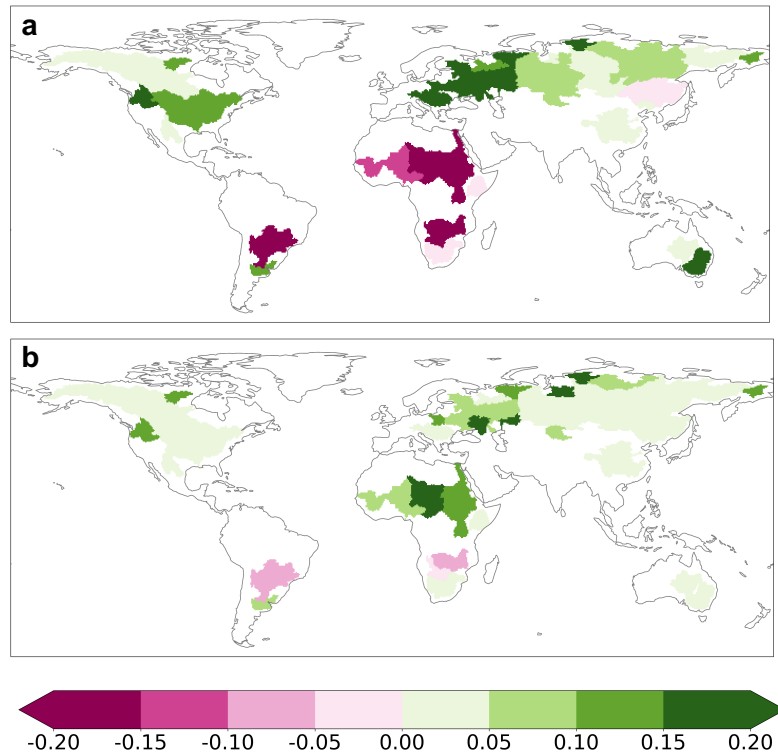

**Figure 7.** Predictive skill differences for basin ET anomalies. (a) The NSE difference between $HydroModel(S_r^{GRACE/FO})$ and $HydroModel(S_r^{RD \times WHC})$. (b) The NSE difference between $HydroModel(S_r^{GRACE/FO})$ and $HydroModel(S_r^{accum})$. White spaces on land represent basins where no model achieves a positive NSE value or no ET data is available.

### 3.4 Linking $S_r$ to vegetation growth

We evaluated the relationship between $S_r^{GRACE/FO}$ and $GPP_{max}$ to link root-zone water storage capacity to vegetation growth. We observed a consistent increase in $S_r^{GRACE/FO}$ alongside $GPP_{max}$ across space (Fig. 8a). This trend reflects the intrinsic relationship between vegetation productivity and water supply across space (Huxman et al., 2004; Ponce-Campos et al., 2013; Hsu et al., 2012). However, we noted a saturation effect at higher $S_r^{GRACE/FO}$ values, suggesting a diminishing influence of water supply beyond a certain threshold. This aligns with ecological principles, particularly in wetter regions, where factors such as nutrient availability and light intensity may dominate over water availability in constraining $GPP_{max}$ (Huxman et al., 2004; Ponce-Campos et al., 2013; Hsu et al., 2012). Notably, since our $S_r^{GRACE/FO}$ estimate is based on the water balance and does not rely on assumed plant-water relations, this evidence supports the reliability of $S_r^{GRACE/FO}$ and sheds light on the intricate interplay of environmental factors influencing vegetation dynamics across landscapes.

We also evaluated the $S_r$ relationship with $GPP_{max}$ using $S_r^{RD \times WHC}$ and $S_r^{accum}$ (Figs. 8b-c), finding that the overall pattern of the functional relationships is similar to that observed using $S_r^{GRACE/FO}$. Specifically, the $GPP_{max}$ increases with increasing $S_r$ before reaching a plateau or showing a notably smaller change with further increases in $S_r$. However, the



thresholds at which this apparent saturation occurs differ: approximately 400 mm for $S_r^{GRACE/FO}$, 50 mm for $S_r^{RD \times WHC}$, and 150

mm for $S_r^{accum}$. To better understand the appropriate threshold, we compared our observed patterns to those inferred from the

spatiotemporal origin of transpiration estimated by Miguez-Macho and Fan (2021). They used inverse modeling and isotopic

analysis to map the annual contribution of root zone water storage (or total past precipitation) to transpiration on a global scale.

By multiplying their root zone water storage contribution with simulated transpiration, we derived a lower-bound $S_r$ estimate

and compared it to annual transpiration across regions (Fig. 8d). Given the widely reported linear relationship between

transpiration and vegetation growth across regions (Ponce-Campos et al., 2013; Biederman et al., 2016; Cooley et al., 2022),

Fig. 8d indicates that the deceleration in vegetation growth may occur at a lower-bound $S_r$ value of 400 mm. As $S_r$ increases

with higher lower-bound $S_r$ (due to their positive correlations with vegetation growth; Figs. 8a-c vs. 8d), the $S_r$ threshold could

exceed the 400 mm inferred from the lower-bound $S_r$ estimate. This aligns better with the threshold inferred from $S_r^{GRACE/FO}$

but is significantly higher than those inferred from $S_r^{RD \times WHC}$ and $S_r^{accum}$. Therefore, $S_r^{GRACE/FO}$ likely provides a more accurate

reflection of real-word spatial patterns of land water supply on vegetation growth than $S_r^{RD \times WHC}$ and $S_r^{accum}$.

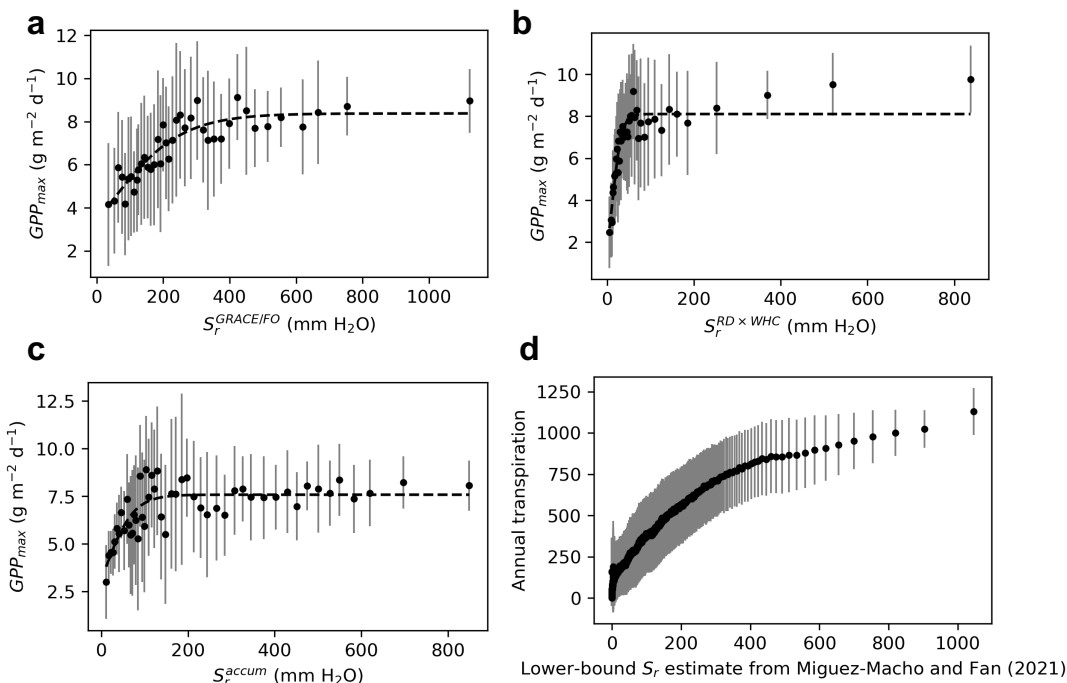

**Figure 8.** Scatterplots of $GPP_{max}$ and $S_r$ across regions based on $S_r^{GRACE/FO}$ (a), $S_r^{RD \times WHC}$ (b), and $S_r^{accum}$ (c). All analyzed
mascons are grouped into 40 equal-sized bins based on $S_r$. Circle and error bar denote the mean and standard deviation of
$GPP_{max}$ within each bin, respectively. The dashed black line in each plot represents a model fit using a nonlinear concave-
down model. (d) is the lower-bound estimate of $S_r$ derived from Miguez-Macho and Fan (2021) in relation to their simulated
annual transpiration. Due to the high resolution of their inverse modeling (30′), model grid cells are grouped into 1000 equal-
sized bins based on the lower-bound estimate of $S_r$. Circle and error bar denote the mean and standard deviation of annual
transpiration within each bin.



## 4 Discussion


Our $S_r^{GRACE/FO}$ estimate provides a conservative lower bound on $S_r$ because the largest TWS drawdown during the GRACE/FO record period may not cover a period during which ET from storage exhausts the entire root-zone water storage capacity, particularly in areas experiencing water accumulation in the root zone due to increased precipitation. This likely explains why our $S_r^{GRACE/FO}$ estimate is lower than $S_r^{accum}$ in North and East Africa, where strong increasing TWS trends were

observed (Fig. 3b and Fig. A2). Additionally, our approach to account for groundwater pumping and surface water may overestimate these signals' actual magnitudes and thus likely contribute to underestimating $S_r$. Specifically, we assumed all negative TWS trends to be caused by groundwater withdrawal and removed them from $S_r^{GRACE/FO}$. However, groundwater withdrawal is concentrated in specific regions such as northwest India, California's Central Valley, and the North China Plain (Rodell et al., 2009; Feng et al., 2013; Liu et al., 2022). Consequently, we may remove TWS depletion trends caused by natural

variability, as seen in the drought-stricken Southeast Brazil (Rodell et al., 2018). This likely explains why $S_r^{GRACE/FO}$ is lower than $S_r^{accum}$ there (Fig. 3b). Furthermore, we used total runoff (which includes surface runoff, snowmelt, and groundwater flow) as a proxy to remove surface water storage change from the TWS drawdown. We used total runoff – as opposed to surface runoff alone (Wang et al., 2023) – due to observational data availability, though doing so may lead to an overestimation of surface water storage change and, therefore, an underestimation of $S_r$.

Despite being conservative, $S_r^{GRACE/FO}$ reveals a substantially larger volume of root-zone water storage capacity than $S_r^{accum}$. One reason for this discrepancy may be the lack of interannual storage variability considered in the $S_r^{accum}$ calculation (Stocker et al., 2023). Although Stocker et al. (2023) used a cumulative water deficit approach to infer root-zone water storage drawdown, akin to our TWS drawdown approach, they found that the annual totals of P exceeded those of ET at almost all locations. Because their method resets the calculation whenever accumulated P-ET is positive, this suggests their method

generally was unable to account for carryover storage and multiyear drawdowns of root-zone storage. Our use of GRACE/FO TWS, which allows for multiyear drawdowns, is supported by recent observations (Goulden and Bales, 2019; Mccormick et al., 2021; Pérez-Ruiz et al., 2022; Peterson et al., 2021; Scott and Biederman, 2019) and modeling efforts (Miguez-Macho and Fan, 2021; Livneh and Hoerling, 2016) suggesting widespread carryover storage effects. Our calculations of $S_r^{GRACE/FO}$ found that the largest TWS drawdown period lasted a median of 2.8 years, with an interquartile range between 1.6 and 5.2 years (Fig.

2c). Even the second and third-largest TWS drawdowns had a median duration of more than one year globally (Figs. 3c-d). These findings align with the results reported in the previously referenced studies on carryover storage effects.

The $S_r^{RD \times WHC}$ estimate notably falls below both $S_r^{GRACE/FO}$ and $S_r^{accum}$. This discrepancy may be attributed to the $RD \times WHC$ approach ignoring plant access to bedrock moisture and groundwater, which are known to significantly affect ET and thus contribute to $S_r$ (e.g., Fan et al., 2017; Rempe and Dietrich, 2018; Mccormick et al., 2021). Moreover, the $RD \times WHC$

approach lacks consideration for root density and its vertical and lateral distribution, simplifying the root zone's complexity into a single effective rooting depth parameter (Federer et al., 2003; Speich et al., 2018). This parameter tends to be shallower than both the maximum rooting depth (Federer et al., 2003) and the depth that contains the upper 95% of the root biomass





(Yang et al., 2016), although these depths may play a disproportionately important role in ecosystem water uptake (Fan et al., 2017; Jackson et al., 1999; Bachofen et al., 2024). Additionally, when dividing $S_r^{GRACE/FO}$ with the same $WHC$ used in $S_r^{RD \times WHC}$

to calculate effective rooting depth, this depth exceeds 2 m in nearly 50% of global vegetated areas, in contrast to Yang et al.'s (2016) estimate of 10% and Stocker et al.'s (2023) estimate of 37%. These results suggest that the potential for plants to tab into deep water stores is more prevalent than previously understood.

Despite different $S_r$ parameterizations, the USGS hydrological model performs poorly in extremely wet and dry regions, such as the Amazon rainforest and much of Australia (Fig. A3), likely due to a lack of calibration of other parameters

or an overly simplistic representation of key hydrological processes. The model's algorithm aims to meet the potential ET (PET), or the atmospheric demand for water, using precipitation and withdrawals from root-zone water storage (Mccabe and Markstrom, 2007). It uses the Hamon equation (Hamon, 1964) to calculate PET, and previous studies (e.g., Sun et al., 2008; Mccabe et al., 2015) have found that the Hamon coefficient needs to be calibrated to generate realistic ET. However, calibrating the Hamon coefficient could absorb or compensate for the $S_r$ parameterization error, undermining the objectiveness of the

USGS model in evaluating the relative accuracy of the three $S_r$ estimates. In very wet regions, the USGS model often simulates the PET significantly lower than incoming precipitation (Fig. A4). Consequently, the model does not need to tap root zone water storage for ET, resulting in little variability in TWS for these regions (Fig. A4). Conversely, in very dry regions, the USGS model simulates the PET to be notably higher than incoming precipitation most of the time, leaving the root-zone water storage close to zero (Fig. A5). However, large variability in TWS was observed by GRACE/FO for these regions, which is

consistent with other studies indicating strong soil moisture variations (Swann and Koven, 2017; Chen et al., 2014). These results suggest that structural errors or uncertainty of other parameters in the USGS model may outweigh the uncertainty of $S_r$ parameterization in these very wet and dry environments.

This paper demonstrates how GRACE/FO data can be used to constrain vegetation water use patterns. Although observed at a coarse resolution, the $S_r^{GRACE/FO}$ can be used to evaluate high-resolution $S_r$ estimates to ensure consistency and

accuracy across different scales. In addition, our methodology can be applied to downscaled TWS products, leveraging techniques such as data assimilation systems or artificial intelligence (Li et al., 2019; Gou and Soja, 2024), to improve the characterization of $S_r$ and its impact on the water and carbon cycles at a higher spatial resolution.

## 5 Conclusions

We used GRACE/FO to provide a direct observational constraint on root-zone water storage capacity ($S_r$), an essential

yet challenging-to-observe variable. The overall better performance of $HydroModel(S_r^{GRACE/FO})$ in simulating TWS and ET observations and the superior $S_r^{GRACE/FO}$ relationship with $GPP_{max}$ altogether imply that $S_r^{GRACE/FO}$ more accurately reflects the real-word root-zone water storage capacity compared to $S_r^{RD \times WHC}$ and $S_r^{accum}$. These results suggest that $S_r$ is, on average, at least 50% larger than the water deficit-based estimate and by a staggering 380% compared to the rooting depth-based estimate. The underestimations by $S_r^{accum}$ and $S_r^{RD \times WHC}$ exceed the random error of $S_r^{GRACE/FO}$, underscoring the need for continued



refinement and validation of $S_r$. Underestimating $S_r$ may lead to overestimating ecosystem sensitivity to water stress, potentially biasing predictions of future carbon cycle (Ukkola et al., 2021; Giardina et al., 2023). Given the strong coupling between the carbon and water cycles, underestimating $S_r$ may also lead to underestimating ecosystem water consumption and overestimating human-available water resources, particularly during droughts and heat waves, with important implications for water resource planning (Zhao et al., 2022; Mastrotheodoros et al., 2020).






## Appendix A

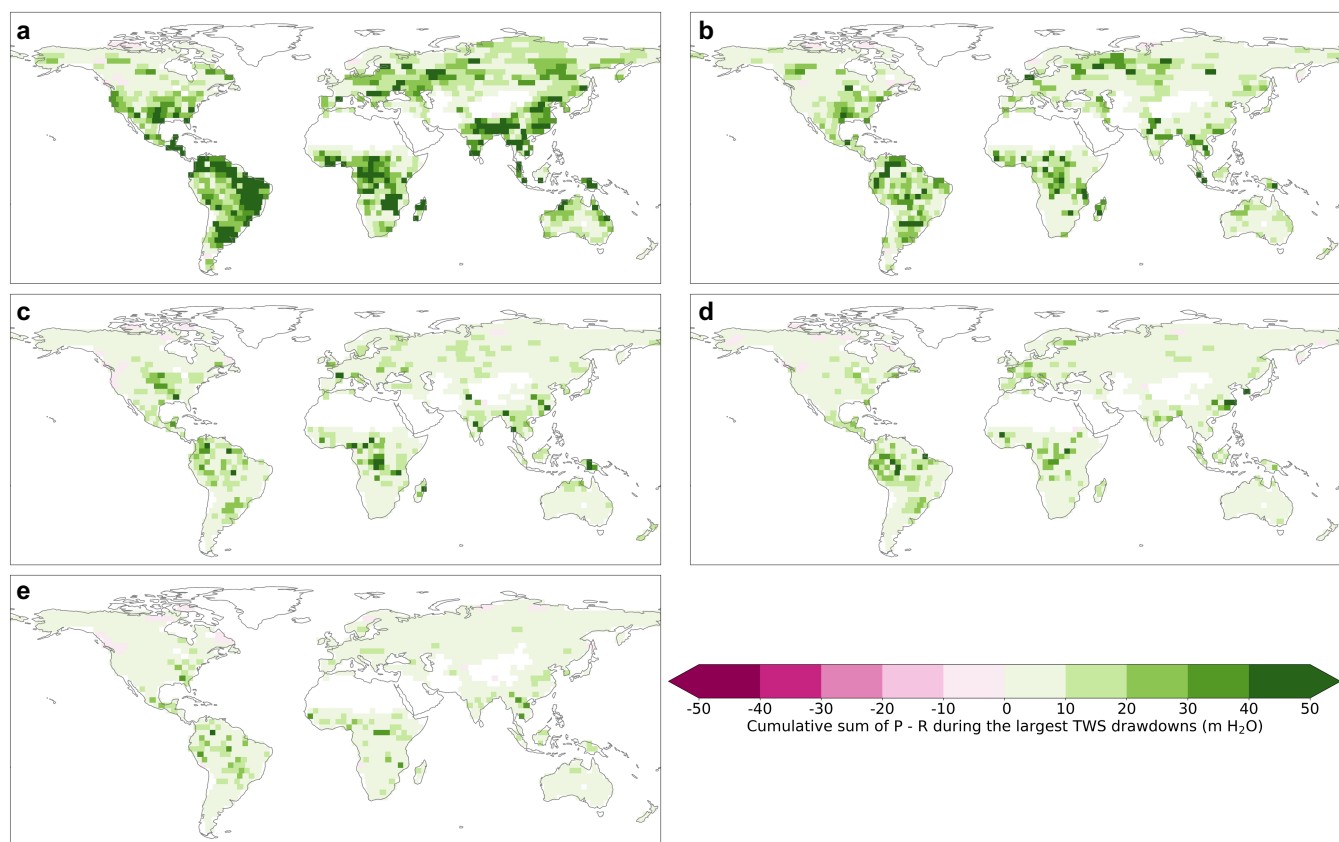

**Figure A1**. The cumulative sum of P - R during the largest (a), the second largest (b), the third largest (c), the fourth largest (d), and the fifth largest (e) TWS drawdowns.




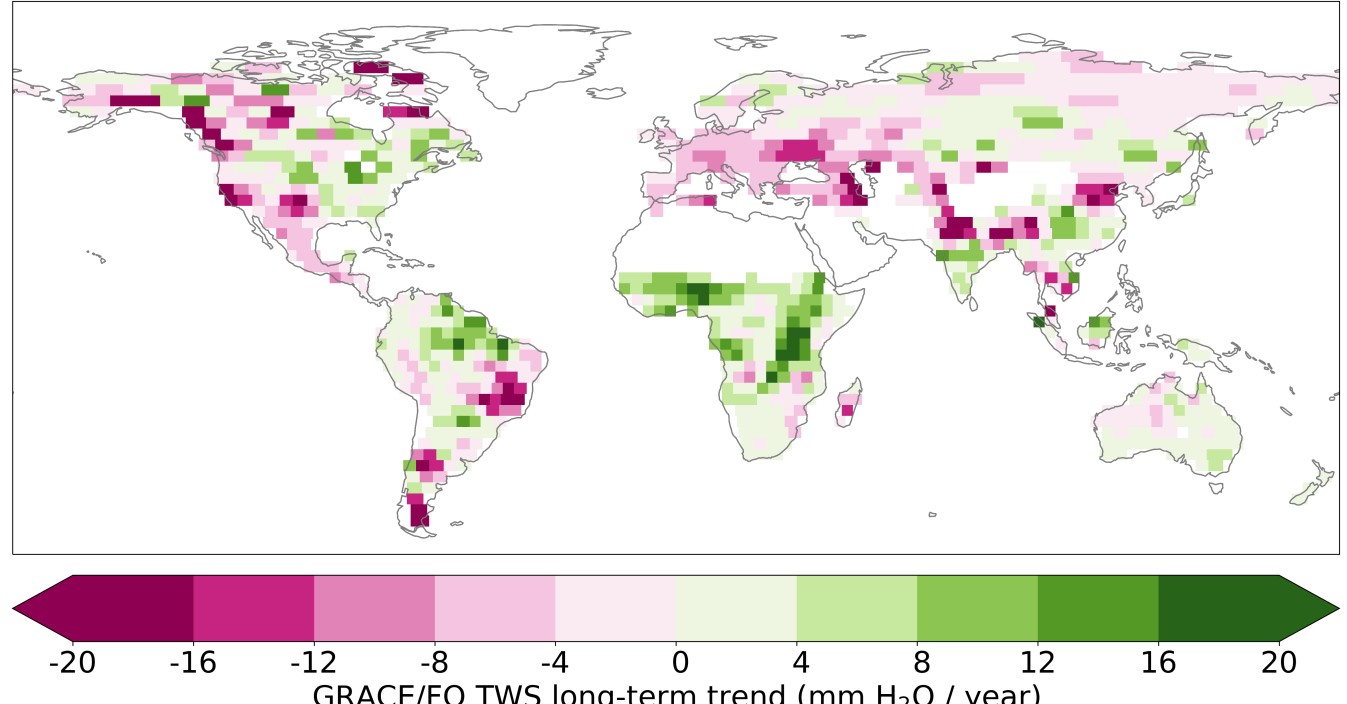

**Figure A2**. Trends in TWS obtained from GRACE/FO observations from 2002 to 2022.



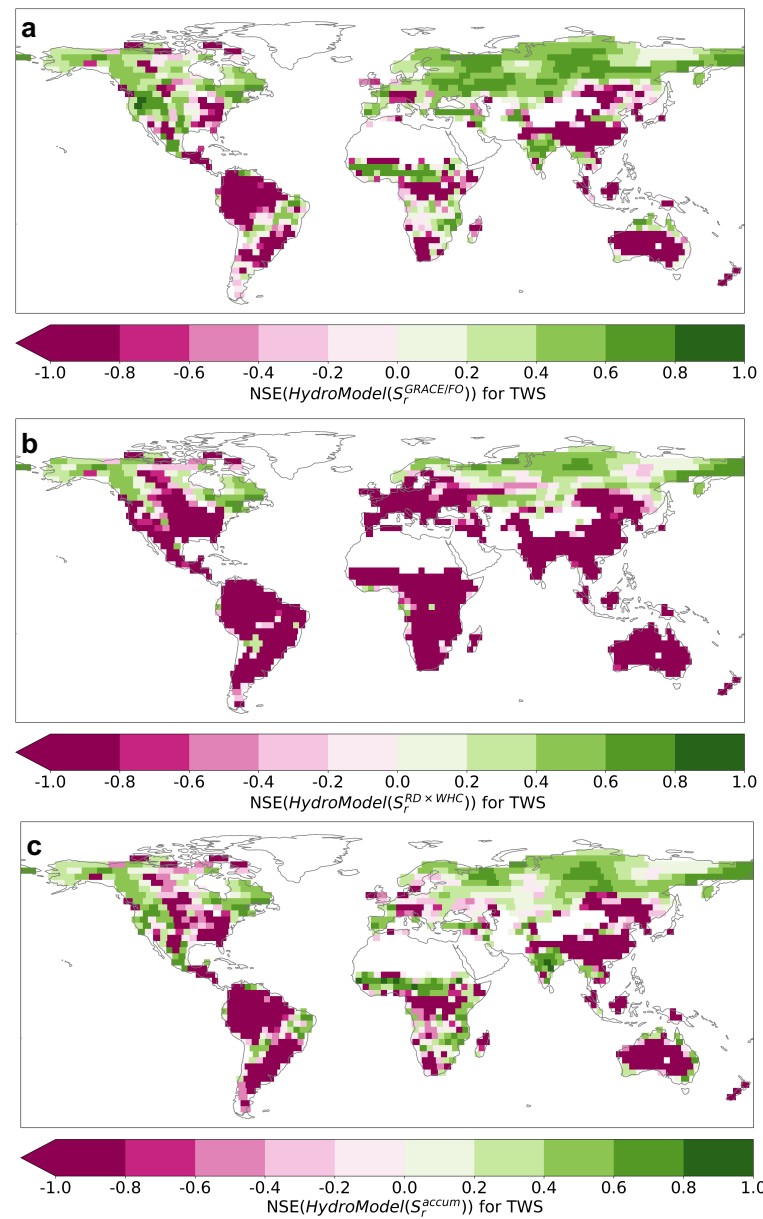

**Figure A3.** NSE values for simulating GRACE/FO TWS by *HydroModel*($S_r^{GRACE/FO}$) (a), *HydroModel*($S_r^{RD×WHC}$) (b), and
*HydroModel*($S_r^{accum}$) (c), respectively.



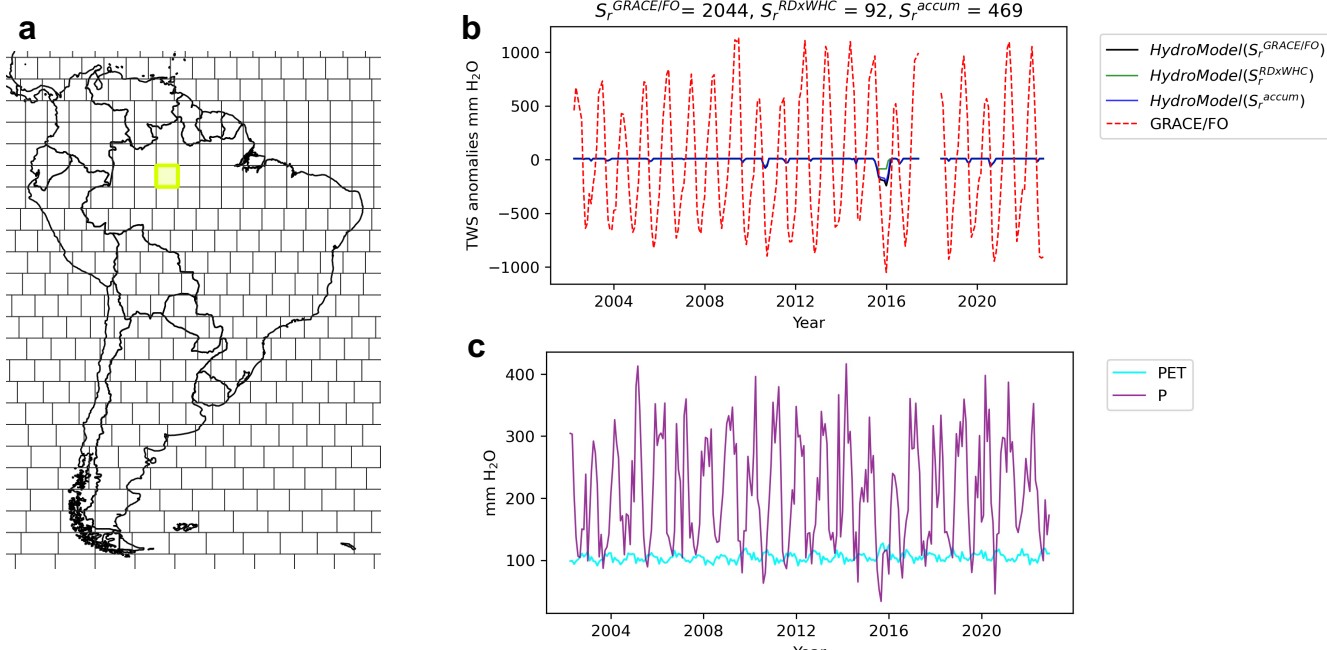

**Figure A4.** Model results for a very wet mascon in the Amazon rainforest (a). (b) The comparison between modeled TWS and GRACE/FO TWS. (c) The comparison between the precipitation (P) forcing and model simulated potential evapotranspiration (PET).




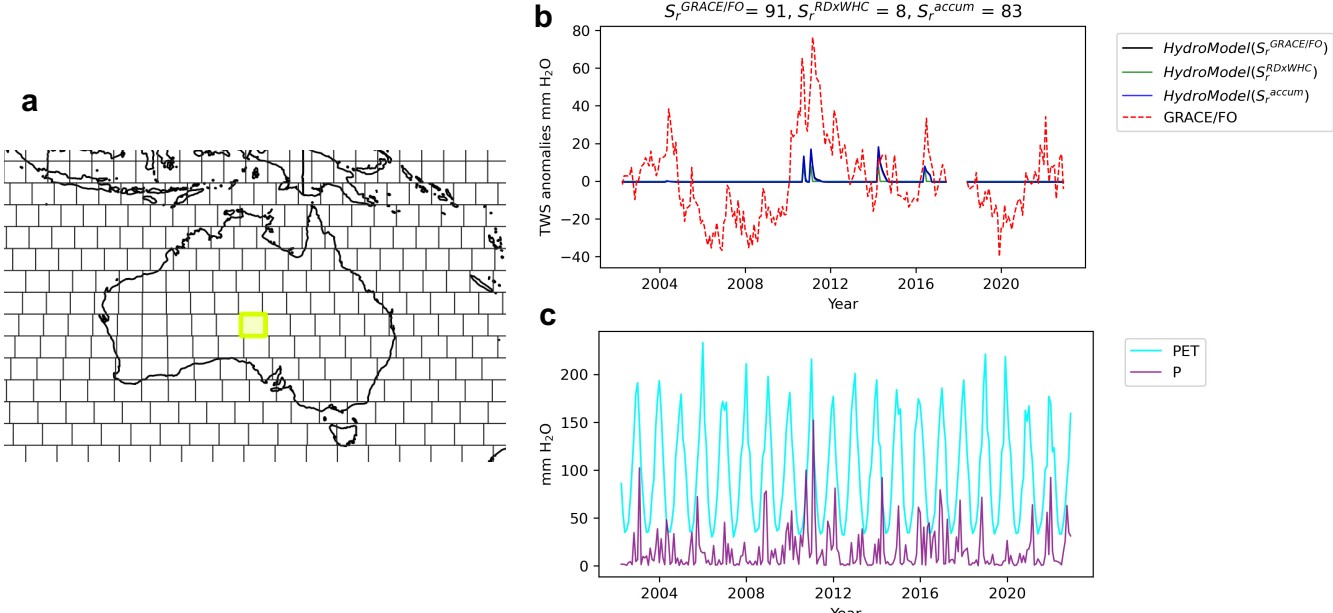

**Figure A5.** Same as Fig. A4 but for a very dry mascon in Australia.



**Code availability**

The working code to retrieve $S_r$ from GRACE/FO is available to reviewers. The final code will be archived on Zenodo upon acceptance of the paper. A DOI link to the archived code will be provided in the final version of the manuscript.

**Data availability**

The $S_r^{GRACE/FO}$ will be archived on Zenodo and a DOI link will be provided in the final version of the manuscript. GRACE and GRACE-FO TWS data are available from the NASA JPL (https://grace.jpl.nasa.gov/data/get-data/jpl_global_mascons/). The
GPCP version 2.3 combined precipitation dataset is available at https://psl.noaa.gov/data/gridded/data.gpcp.html. ERA5 reanalysis is available at https://www.ecmwf.int/en/forecasts/datasets/reanalysis-datasets/era5. MODIS land cover data are available    at    https://lpdaac.usgs.gov/products/mcd12c1v006/.    Water-balance-based    ET    data    is    available    at https://doi.org/10.5281/zenodo.8339655.    G-RUN    global    runoff    reconstruction    data    is    available    at https://figshare.com/articles/dataset/GRUN_Global_Runoff_Reconstruction/9228176.

**Author contribution**

MZ: Conceptualization; Data curation; Formal analysis; Funding acquisition; Methodology; Writing - original draft. ELM: Methodology; Writing - review & editing. GA: Methodology; Writing - review & editing. AGK: Writing - review & editing. BL: Writing - review & editing.

**Competing interest**

The authors declare that they have no conflict of interest.

**Acknowledgments**

This study was funded by the USGS grant G24AP00031 to the University of Idaho. In addition, ELM was funded by the NSF Graduate Research Fellowship program and AGK was funded by the NSF DEB 1942133 and the Alfred P. Solan Foundation.

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
