# Peer review of "Substantial root-zone water storage capacity observed by GRACE"

_EGUsphere, 2024_

## Author Comment (AC1)

In the following responses, reviewers' comments are reproduced in their entirety in black, and the authors' responses are noted in blue.

**Reviewer 1**

Reviewer: Review of "Substantial root-zone water storage capacity observed by GRACE and GRACE/FO" by Zhao et al. . This paper describes the use of TWS estimates from the GRACE satellite project to estimate multi-year water storage changes. Negative changes are used to estimate a lower-bound on root-zone water storage (Sr). The estimates are compared to two alternative (Sr) methodologies, and all three Sr estimates are used to parameterize a hydrologic model. The main result is that the authors' Sr is significantly larger than the previously described Sr estimates.

Comments

General

I found the authors' use of GRACE data to be novel, and the results interesting. The paper is well-written and generally clear.
Response: Thanks for the positive feedback.

As with other GRACE studies, the spatial resolution of the data is relatively coarse, so I suggest adding some discussion of how these results might be applied in the operational configuration of land models, which would typically have finer spatial resolution.
Response: Thanks for your comment and suggestion. In our revised manuscript, we will discuss two ways to use our $S_r$ estimates for land surface modeling. First, $S_r^{GRACE/FO}$ can be used for evaluating default $S_r$ parameterization used in the model at the coarse-scale of GRACE/FO data in conjunction with other analyses. For instance, if a model simulates low ET during droughts in a region where the $S_r$ value is also low compared to $S_r^{GRACE/FO}$, the default value may be increased based on $S_r^{GRACE/FO}$. Second, we will discuss approaches for developing finer-scale GRACE-based $S_r$ products, such as using downscaled TWS products developed through machine learning and data assimilation techniques (Gou & Soja, 2024; Li et al., 2019) .

Reviewer: Abstract
The maximum water held would be the difference from saturation to wilting point. But saturated conditions are unlikely to occur at these spatial scales in many regions.
Response: Our root zone storage capacity includes water uplifted from groundwater, and thus, it is not limited by the wilting point and saturation. In the revision, we will rephrase it to "the maximum amount of water available for plant uptake."

Reviewer: 1st sentence defines Sr, and the next sentence discusses simulations. Perhaps add a sentence indicating how Sr is used in a modeling context after the 1st sentence to provide context.
Response: Thanks. We will add a couple of sentences to explain why and how $S_r$ is relevant to model simulation.

Reviewer: Line 15: to be clear, GRACE measures gravity and TWS is inferred from that, so the use of the word 'direct' can be problematic. There are other geophysical processes that affect time-varying gravity.
Response: We will remove 'direct.'

Reviewer: Line 20: what does 'correlates realistically' mean? Can you use a more specific or quantitative description?
Response: We will specify the relationship as a saturating relationship in the revision.

Reviewer: Introduction
Line 26: 'plants can store during wet periods' should be 'plants can access'? i.e. plants aren't storing the water, the soil is storing water.
Response: We will revise the sentence to "the more water plant roots can access for use in droughts."

Reviewer: Line 37: why would it overlook rock moisture and groundwater? This sentence implies a different reason besides uncertainties in rooting depths or hydraulic properties, which are mentioned previously.
Response: Thank you for this comment. The reason water stored in weathered rocks and groundwater is often overlooked is that most approaches typically set rooting depth shorter than simulated soil thickness and assume that roots do not extend into deeper unsaturated zones. However, recent studies have shown that this assumption is not always accurate (Rempe and Dietrich, 2018; Fan et al., 2017). In fact, in many ecosystems, plant roots can penetrate beyond the shallow soil layer into weathered bedrocks to access deep water storage, including groundwater, especially during dry seasons and droughts (Mccormick et al., 2021; Maxwell and Condon, 2016).

We will clarify this and highlight the importance of bedrock moisture and groundwater in our definition of root-zone storage capacity in the revised manuscript.

Reviewer: Line 49: again, the word 'direct' I find problematic. If you wish to use this word, perhaps add a sentence explaining its use.
Response: We will remove 'direct.'

Reviewer: Methods

Line 61: clearly, 'root-zone' implies vegetated areas, but what might one learn from this method in more arid regions?

Response: Thank you for your question. In more arid regions such as deserts, our approach may capture moisture storage capacity for bare soil evaporation. We will clarify this point in the revision.

Reviewer: Line 68: typically P, ET, and R refer to fluxes. To be more consistent with other literature, consider using rate or flux units consistently and include a summation symbol in equation 1.

Response: We will use flux units and add a summation symbol in equation 1 in the revision.

Reviewer: Line 75: 'consumed' could be changed to 'transpired' or 'returned to the atmosphere'

Response: We will change it to 'transpired.'

Reviewer: Line 82: in areas in which widespread groundwater use is absent, how will this trend removal affect your results? Is it likely to increase or decrease your Sr estimates for such areas? Could you use maps of irrigated area, such as AQUASTAT, to confine this operation to areas where widespread irrigation occurs?

Response: You raised a good point here. In some cases, long-term trends in TWS can be associated with precipitation trends in responses to climate change. In those cases, removing long-term linear trends likely leads to underestimation of $S_r$. However, we found that regions showing significant TWS decreasing trends largely coincide with known irrigation areas identified in AQUASTAT data, except in some high Arctic locations (Figs. RC1_1a, b). Thus, our $S_r$ estimates may be underestimated in these high Arctic regions. We will discuss this limitation in the revision.

The AQUASTAT dataset has its own uncertainties and limitations. For instance, it is based on statistics from 2000-2008 and is particularly uncertain in high-latitude regions (Fig. RC1_1c). Consequently, it may not provide reliable information on groundwater use in some areas of the globe.

[Figure]

**Figure RC1_1.** (a) Trends in TWS obtained from GRACE/FO observations from 2002 to 2022. (b) Percentage of area equipped for irrigation that is actually irrigated. (c) Map quality marks assigned to each country for area equipped for irrigation in (b). (b-c) are from the Global Map of Irrigation Areas ⁻ version 5.0 by AQUASTAT available at https://firebasestorage.googleapis.com/v0/b/fao-aquastat.appspot.com/o/PDF%2FMAPS%2Fgmia_v5_lowres.pdf?alt=media&token=d098a48f-ab49-4eae-a16e-82a5779f924e

Reviewer: Line 91: how runoff is used here is not clear to me. Is there a budget equation that could be shown? What does 'surface water' encompass; rivers, lakes, reservoirs, ...?

Response: Yes, surface water here encompasses water stored in rivers, lakes, and reservoirs. In GRACE/FO TWS decomposition studies (Bhanja et al., 2016; Getirana et al., 2017; Shamsudduha & Taylor, 2020; Thomas et al., 2017; Wang et al., 2023), surface runoff ($Q$) is commonly used as a proxy for surface water storage change ($\Delta SW$), expressed as $\Delta SW = Q$. This approach assumes $Q$ directly contributes to an increase in surface water levels within the drainage network. This approach also assumes that it takes approximately one month for $Q$ to exit the drainage system, aligning with the monthly time step of GRACE/FO data.

In our study, we used total runoff ($R$), which includes both surface runoff ($Q$) and subsurface runoff, as a proxy for $\Delta SW$ (i.e., $\Delta SW = R$), as subsurface runoff which is groundwater discharge to rivers also contributes to surface water storage changes.

We will clarify the methodology and justification further in the revised manuscript and include the water budget equation ($\Delta SW = R$) for clarity.

Reviewer: Line 109: to what extent is Yang 2016 a model-based dataset versus an observational dataset?

Response: Yang et al. (2016) is a fully model-based dataset. It relies on Guswa (2008)'s analytical model that estimates rooting depth, which balances the carbon gain and cost of any additional roots. While such model-based datasets are valuable for providing comprehensive coverage and insights into complex processes, they do not incorporate direct observational data for validation or correction. We will discuss this caveat in the revision.

Reviewer: Line 111: how is water holding capacity defined? Field capacity minus wilting point?

Response: The reviewer is correct. Field capacity is defined as the difference between field capacity and permanent wilting point. We will add this definition in our revised manuscript.

Reviewer: Line 132: why is this an approximation? Are there other modeled water storage components in HydroModel that were ignored?

Response: There are no other modeled water storage components in the USGS model. We will clarify this in the revision.

Reviewer: Line 141: 'ET anomalies'

Response: We will correct it in the revised manuscript.

Reviewer: Line 146: Does Xiong 2023 use GRACE water storage for their ET estimates? If so, does that reduce its independence from your results?

Response: Yes, Xiong et al. (2023) used GRACE for their ET estimates. In the revised manuscript, we will replace Xiong et al. (2023) ET estimates with the latest version (v4.1) of the Global Land Evaporation Amsterdam Model (GLEAM) ET dataset (https://www.gleam.eu/) to validate our model results. The GLEAM ET is an improved dataset, addressing key shortcomings present in other gridded ET products. For example, it combines hybrid learning from eddy-covariance and sap flow to capture vegetation responses to drought more accurately (Koppa et al., 2022), and it explicitly accounts for plant access to groundwater (Hulsman et al., 2023). Importantly, the GLEAM ET is independent of GRACE/FO and, therefore, allows robust validation that is free from circularity.

Reviewer: Line 169: you say that you compare the two datasets, but you don't explicitly say what your hypothesized relationship between them is, so the justification here seems weak. In areas that are not water limited, one could imagine that GPP would be high, but a deep root zone is not necessary. Perhaps expand further on your reasoning in this paragraph.

Response: Thank you for this suggestion. We will revise this paragraph as follows to justify the GPP analysis in the revised manuscript (new text in bold).

*"The $S_r^{GRACE/FO}$ is derived from the water balance, but its ecological relevance remains undetermined. **We hypothesize that ecosystems with higher biomass, particularly in wetter regions, develop larger $S_r$ as a long-term adaptation to manage periodic water surpluses, promoting water retention and maintaining productivity during dry periods. Even in regions that are not water-limited, a larger $S_r$ may still serve as a buffer against interannual precipitation variability, thus contributing to sustained productivity despite the less direct influence of immediate water limitations. To investigate this hypothesis, we compared $S_r^{GRACE/FO}$ with an independent measure of ecosystem productivity**..."*

Reviewer: Line 194: is this saying that the durations shown in 3c) and 3d) are often larger than that shown in 2b)?

Response: No. The average duration of the first, second, and third-largest TWS drawdowns are 2.8, 1.6, and 1.2 years, respectively. This indicates that the durations in Fig.3 c) and d) are often shorter than those shown in Fig. 2b.

Reviewer: Line 225: Do these patterns correlate with a particular land cover or vegetation type?

Response: We did not find a clear correlation with a particular land cover or vegetation type. This may be due to the large spatial resolution of GRACE/FO data, which represents combined signals from various land cover types within its 3° × 3° footprint. As

a result, it is challenging to isolate patterns specific to individual land cover or vegetation types.

Reviewer: Line 271: plot d) is unclear to me.  You create an Sr estimate from Miguez-Macho 2021, but then plot it against transpiration instead of GPP; why is this done differently from a) - c)?
Response: Thank you for pointing this out. The reason for this difference is that Miguez-Macho and Fan (2021) only provided transpiration data, not GPP. To maintain consistency across all plots a) – d) in our revised manuscript, we will convert their transpiration data to GPP. This conversion will be done using a linear regression ($R^2$ = 0.86) between their transpiration data and the GPP derived from MODIS and FLUXNET. This will ensure that plot d) aligns with plots a) – c) in its use of GPP, addressing the inconsistency you noted. The new figure will be as follows:

[Figure]

Reviewer: Figure 8: why are the x- and y-axis ranges different for plots a) - c)?  It is harder to compare the scatterplots because of this.
Response: We will adjust plots a) – d) to make x-axis and y-axis ranges identical for all subplots (see the new figure above).

Reviewer: Discussion

Line 321: does root-accessible water require that the roots physically occupy the entire storage domain? For example, as soils dry, upward moisture fluxes can occur which might replenish soil moisture deficits near roots. Might this help explain the mismatch between observed rooting depths and the Sr estimates here?
Response: Thank you for this insightful comment. You are correct that roots do not necessarily need to physically occupy the entire storage domain. Processes such as the capillary force can indeed move deep water storage upward to replenish moisture near the roots, especially during dry conditions. Such mechanism could be the reason for the observed differences between the rooting depth-based estimation and our GRACE/FO-based estimation. We will include this discussion in the revised manuscript.

Reviewer: Line 325: one could also interpret your Sr/WHC as simply the effective soil depth. For land models that do not use an explicit Sr variable, this could indicate that models with a soil depth < 2m (i.e. some of the GLDAS models) are likely incapable of simulating these kinds of drawdowns, which would have implications for studies of groundwater that have used GLDAS to remove the soil moisture component of TWS.
Response: Agreed. We will discuss these implications in the revised manuscript.

Reviewer: Line 326: 'tap'
Response: Corrected. Thank you.

Reviewer: Figure A1: how does this result relate to the relationship between magnitude and duration? Does it imply that during the largest drawdowns, there is also the largest 'net precipitation'? That seems counterintuitive.
Response: Thank you for your observation. Figure A1 currently shows the cumulative sum of P – R during the drawdown periods. The largest drawdown does indeed correspond to the longest duration, which results in a higher cumulative sum of P – R. We recognize that this might seem counterintuitive, as it suggests that the largest drawdowns also have the largest 'net precipitation.' To clarify this, we will revise the figure to present the average P – R instead of the *cumulative* sum. This adjustment will remove the influence of duration and reflect the mean P – R during the drawdown periods.

**References**

Bhanja, S. N., Mukherjee, A., Saha, D., Velicogna, I., & Famiglietti, J. S. (2016). Validation of GRACE based groundwater storage anomaly using in-situ groundwater level measurements in India. *J. Hydrol., 543*, 729-738.

Getirana, A., Kumar, S., Girotto, M., & Rodell, M. (2017). Rivers and Floodplains as Key Components of Global Terrestrial Water Storage Variability. *Geophysical Research Letters, 44*(20), 10,359-310,368. https://agupubs.onlinelibrary.wiley.com/doi/abs/10.1002/2017GL074684

Gou, J., & Soja, B. (2024). Global high-resolution total water storage anomalies from self-supervised data assimilation using deep learning algorithms. *Nature Water, 2*(2), 139-150. https://doi.org/10.1038/s44221-024-00194-w

Guswa, A. J. (2008). The influence of climate on root depth: A carbon cost-benefit analysis. *Water Resources Research, 44*(2). https://agupubs.onlinelibrary.wiley.com/doi/abs/10.1029/2007WR006384

Hulsman, P., Keune, J., Koppa, A., Schellekens, J., & Miralles, D. G. (2023). Incorporating Plant Access to Groundwater in Existing Global, Satellite-Based Evaporation Estimates. *Water Resources Research, 59*(8), e2022WR033731. https://agupubs.onlinelibrary.wiley.com/doi/abs/10.1029/2022WR033731

Koppa, A., Rains, D., Hulsman, P., Poyatos, R., & Miralles, D. G. (2022). A deep learning-based hybrid model of global terrestrial evaporation. *Nature Communications, 13*(1), 1912. https://doi.org/10.1038/s41467-022-29543-7

Li, B., Rodell, M., Kumar, S., Beaudoing, H. K., Getirana, A., Zaitchik, B. F., et al. (2019). Global GRACE data assimilation for groundwater and drought monitoring: Advances and challenges. *Water Resources Research, 55*(9), 7564-7586.

Shamsudduha, M., & Taylor, R. G. (2020). Groundwater storage dynamics in the world's large aquifer systems from GRACE: Uncertainty and role of extreme precipitation. *Earth Syst. Dyn., 11*, 755-774.

Thomas, B. F., Caineta, J., & Nanteza, J. (2017). Global assessment of groundwater sustainability based on storage anomalies. *Geophys. Res. Lett., 44*, 11 445-411 455.

Wang, S., Li, J., & Russell, H. A. (2023). Methods for estimating surface water storage changes and their evaluations. *Journal of Hydrometeorology, 24*(3), 445-461.

Yang, Y., Donohue, R. J., & McVicar, T. R. (2016). Global estimation of effective plant rooting depth: Implications for hydrological modeling. *Water Resources Research, 52*(10), 8260-8276. https://agupubs.onlinelibrary.wiley.com/doi/abs/10.1002/2016WR019392

---

## Author Comment (AC2)

In the following responses, reviewers' comments are reproduced in their entirety in black, and the authors' responses are noted in blue.

**Reviewer 2**

Reviewer: In this work, the authors use the Gravity Recovery and Climate Experiment (GRACE) and GRACE Follow-on (FO) to estimate root-zone storage capacity (Sr). They find estimates of Sr that are much larger than those using mass-balance approaches and rooting depth parameterizations. I found the work interesting, and the writing was succinct and clear. However, I had a difficult time understanding the assumptions and the implications of these assumptions to evaluate the results. I think the authors need to be much clearer about the implications of their assumptions.
Response: Thanks for your overall positive comment.

Main comments:
- Reviewer: The proposed method is quite different from previous work because it directly uses total water storage (TWS) from GRACE. However, GRACE measures a combination of surface water, groundwater, soil moisture, snow and ice. You explain how you remove the streamflow and snow/ice…but how do you remove the effect of groundwater? Are you assuming that groundwater is part of Sr? In some cases, as water table becomes more shallow, conditions become anoxic for plants…wouldn't this decrease Sr? The role of gw in Sr calculations must be better explained and the assumptions clearly laid out.
  Response: Thank you for this comment. Natural groundwater variability is indeed included in our definition of root-zone water storage capacity ($S_r$), and we provide clarification below. As Reviewer#1 correctly pointed out, root-accessible water does not require roots to physically occupy the entire storage domain. Processes such as the capillary rise can move deep water upward to the root zone for vegetation transpiration, especially during dry seasons and droughts. Many studies have shown that natural groundwater variability (such as its seasonal variation) strongly correlates with precipitation minus evapotranspiration (e.g., Li et al., 2015).

  Including groundwater in the calculation of $S_r$ broadens the traditional definition of the "root zone," which is typically confined to the unsaturated soil layer, by recognizing the fact that the root zone is dynamic and access deep groundwater and bedrock moisture during prolonged droughts and high transpiration demand (Gao et al., 2024). Several recent studies (McCormick et al., 2021; Singh et al., 2020; Stocker et al., 2023) have also included groundwater in their definitions of $S_r$. This inclusion is well-supported by recent studies based on *in situ* groundwater (Baldocchi et al., 2021; Fan et al., 2017; Li et al., 2015; Thompson et al., 2011), remote sensing observations (Koirala et al., 2017; Rohde et al., 2024), and modeling efforts (Hain et al., 2015; Miguez-Macho & Fan, 2021), all of which showed that groundwater significantly contributes to ET and is accessible to plants, especially during extreme droughts.

In many ecosystems, water stress can stimulate root growth into deep subsurface through the capillary rise effect, with roots extending to the capillary fringe and the water table, as observed in both field and laboratory studies (Fan et al., 2017; Kuzyakov & Razavi, 2019; Naumburg et al., 2005; Orellana et al., 2012). Although individual shallow-rooted plants (e.g., grassland sites) may not directly tap into groundwater, the large spatial scale of GRACE/FO likely captures water uptake across diverse vegetation types. This blending makes it likely that vegetation types not typically associated with groundwater use may still access it indirectly, such as through hydraulic redistribution by neighboring deeper-rooted plants (e.g., Espeleta et al., 2004; Orellana et al., 2012). Indeed, satellite observations have revealed widespread plant-groundwater interactions at large spatial scales (Koirala et al., 2017), even in dryland regions dominated by grasslands (Rohde et al., 2024; Wang et al., 2023).

Neglecting groundwater in root zone storage capacity can lead to underestimation of land and air interactions (Dong et al., 2022; Maxwell & Condon, 2016; Schlemmer et al., 2018), affect accurate runoff simulation (Hahm et al., 2019), and misrepresent vegetation resilience to droughts and heat waves (Esteban et al., 2021; Jiménez-Rodríguez et al., 2022).

Overall, our $S_r^{GRACE/FO}$ definition aligns with our comparison dataset $S_r^{accum}$ from Stocker et al. (2023) and helps explain why the traditional rooting depth approach ($S_r^{RD \times WHC}$), which does not include groundwater, yields lower values than $S_r^{GRACE/FO}$ and $S_r^{accum}$. This expanded definition is also supported by the latest research on groundwater-vegetation interactions. We will add these discussions to the revised manuscript.

- Reviewer: The proposed method is also quite different from previous methods in the spatial and temporal scale. You are looking at monthly data at 3x3 degrees. This would include several ecosystems that behave very differently. It also includes multi-year droughts…whereas other calculations would account for periods of deficit (calculated at the daily timescale) with a certain return period. This is a completely different metric…is it really appropriate to compare these?
  Response: We appreciate the reviewer's comment. Indeed, our method is fundamentally different from previous approaches. However, we contend that the comparability of the two metrics rests on their shared definition of the physical processes involved. Both $S_r^{accum}$ from Stocker et al. (2023) and $S_r^{GRACE/FO}$ define root zone storage capacity in an identical manner, encompassing groundwater and bedrock moisture and averaging across diverse ecosystems at large spatial scales.

- Reviewer: I am having a difficult time understanding physically what it means to calculate deltaTWS as the difference between TWS anomalies. Are you assuming that the soil will be at saturation at the beginning of the drawdawn, but

will never reach saturation throughout the drawdawn period? Is this an appropriate assumption?

Response: No, our method does not assume soil saturation at the beginning of the drawdown. In fact, saturation is unlikely to occur at the spatial and temporal scales of $S_r^{GRACE/FO}$. During the drawdown period, deltaTWS represents the water, in equivalent water heights (mm), that an ecosystem has used for ET consumption beyond what is available from effective precipitation (precipitation minus total runoff). This calculation does not require saturation and provides only a lower-bound estimate of the root zone storage capacity which must exist in order to explain ET patterns.

- Reviewer: I don't think you should use GRACE to evaluate the performance of HydrModel that includes GRACE information. You state that this is not circular…but it is. Another metric could be streamflow, it would be independent.
Response: Agreed. In the revised manuscript, we will evaluate the model performance with the latest version (v4.1) of the Global Land Evaporation Amsterdam Model (GLEAM) ET dataset (https://www.gleam.eu/). The GLEAM ET is an improved dataset, addressing key issues present in other gridded ET products. For example, it combines hybrid learning from eddy-covariance and sap flow to capture vegetation response to drought more accurately (Koppa et al., 2022), and it explicitly accounts for plant access to groundwater (Hulsman et al., 2023). Importantly, the GLEAM ET is independent of GRACE/FO and, therefore, allows robust validation that is free from circularity.

   We appreciate your suggestion to use streamflow for model evaluation. Unfortunately, streamflow is not the most reliable measure for evaluating the USGS model. First, the USGS model primarily parameterizes streamflow based on precipitation, with subsurface storage contributing only when the storage "bucket" is full (McCabe & Markstrom, 2007). This oversimplified scheme does not adequately represent base flow, which is more directly influenced by water stored in the subsurface including groundwater (Reager et al., 2014). Second, the two key parameters governing streamflow generation – the fraction of precipitation converted to direct runoff and the fraction of spillover from the storage bucket converted to runoff – are globally uniform and not calibrated to local conditions. This lack of calibration limits the model's capability to capture spatial and temporal variability in streamflow dynamics. Third and more importantly, compared to precipitation, ET, and TWS anomalies, streamflow is the smallest component of the Earth's hydrological cycle. As a result, it is less sensitive to $S_r$ parameterizations. For these reasons, we will use ET as an evaluation metric following Wang-Erlandsson et al. (2016) in the revised manuscript.

- Reviewer: The part about linking Sr to vegetation growth was not very convincing. I think you are comparing maximum GPP to the point of saturation…so if I understand correctly what you are showing is that vegetation

activity is enhanced when there is enough water. I don't think this argument is necessary for your paper.

Response: We appreciate the reviewer's comment regarding the linkage between soil moisture storage ($S_r$) and vegetation growth. (Huxman et al., 2004; Ponce-Campos et al., 2013) We agree that vegetation productivity is often determined by water availability. However, it is insightful to examine the specific role of $S_r$ in influencing vegetation growth.

As Reviewer#1 commented, it might seem intuitive to assume that in regions with abundant precipitation, GPP would be high, and a large $S_r$ might be unnecessary. However, our analysis shows that a large $S_r$ is still essential in these ecosystems, suggesting vegetation growth is not solely determined by the immediate availability of water but also by the ecosystem's capacity to store it. Therefore, our analysis can reveal how $S_r$ modulates plant-water interactions across diverse hydroclimatic conditions.

Furthermore, comparing $GPP_{max}$ to the point of saturation provides an independent assessment of the relative accuracy of the three $S_r$ products, offering additional insights beyond those derived from the USGS models.

In light of these discussions, we believe that exploring the relationship between $S_r$ and vegetation growth provides useful information. In the revised manuscript, we will further clarify and elaborate the rationale for linking $S_r$ to vegetation growth to ensure that the relevance and importance of this analysis are clear.

**References**

Baldocchi, D., Ma, S., & Verfaillie, J. (2021). On the inter- and intra-annual variability of ecosystem evapotranspiration and water use efficiency of an oak savanna and annual grassland subjected to booms and busts in rainfall. *Global Change Biology, 27*(2), 359-375. https://onlinelibrary.wiley.com/doi/abs/10.1111/gcb.15414

Dong, J., Lei, F., & Crow, W. T. (2022). Land transpiration-evaporation partitioning errors responsible for modeled summertime warm bias in the central United States. *Nature Communications, 13*(1), 336. https://doi.org/10.1038/s41467-021-27938-6

Espeleta, J. F., West, J. B., & Donovan, L. A. (2004). Species-specific patterns of hydraulic lift in co-occurring adult trees and grasses in a sandhill community. *Oecologia, 138*(3), 341-349. https://doi.org/10.1007/s00442-003-1460-8

Esteban, E. J. L., Castilho, C. V., Melgaço, K. L., & Costa, F. R. C. (2021). The other side of droughts: wet extremes and topography as buffers of negative drought effects in an Amazonian forest. *New Phytologist, 229*(4), 1995-2006. https://nph.onlinelibrary.wiley.com/doi/abs/10.1111/nph.17005

Fan, Y., Miguez-Macho, G., Jobbágy, E. G., Jackson, R. B., & Otero-Casal, C. (2017). Hydrologic regulation of plant rooting depth. *Proceedings of the National*

Academy of Sciences, 114(40), 10572-10577.
https://www.pnas.org/content/pnas/114/40/10572.full.pdf

Gao, H., Hrachowitz, M., Wang-Erlandsson, L., Fenicia, F., Xi, Q., Xia, J., et al. (2024). Root zone in the Earth system. *EGUsphere, 2024*, 1-30.
https://egusphere.copernicus.org/preprints/2024/egusphere-2024-332/

Hahm, W. J., Dralle, D. N., Rempe, D. M., Bryk, A. B., Thompson, S. E., Dawson, T. E., & Dietrich, W. E. (2019). Low Subsurface Water Storage Capacity Relative to Annual Rainfall Decouples Mediterranean Plant Productivity and Water Use From Rainfall Variability. *Geophysical Research Letters, 46*(12), 6544-6553.
https://agupubs.onlinelibrary.wiley.com/doi/abs/10.1029/2019GL083294

Hain, C. R., Crow, W. T., Anderson, M. C., & Yilmaz, M. T. (2015). Diagnosing Neglected Soil Moisture Source–Sink Processes via a Thermal Infrared–Based Two-Source Energy Balance Model. *Journal of Hydrometeorology, 16*(3), 1070-1086. https://journals.ametsoc.org/view/journals/hydr/16/3/jhm-d-14-0017_1.xml

Hulsman, P., Keune, J., Koppa, A., Schellekens, J., & Miralles, D. G. (2023). Incorporating Plant Access to Groundwater in Existing Global, Satellite-Based Evaporation Estimates. *Water Resources Research, 59*(8), e2022WR033731.
https://agupubs.onlinelibrary.wiley.com/doi/abs/10.1029/2022WR033731

Huxman, T. E., Smith, M. D., Fay, P. A., Knapp, A. K., Shaw, M. R., Loik, M. E., et al. (2004). Convergence across biomes to a common rain-use efficiency. *Nature, 429*(6992), 651-654. https://doi.org/10.1038/nature02561

Jiménez-Rodríguez, C. D., Sulis, M., & Schymanski, S. (2022). Exploring the role of bedrock representation on plant transpiration response during dry periods at four forested sites in Europe. *Biogeosciences, 19*(14), 3395-3423.

Koirala, S., Jung, M., Reichstein, M., de Graaf, I. E. M., Camps-Valls, G., Ichii, K., et al. (2017). Global distribution of groundwater-vegetation spatial covariation. *Geophysical Research Letters, 44*(9), 4134-4142.
https://agupubs.onlinelibrary.wiley.com/doi/abs/10.1002/2017GL072885

Koppa, A., Rains, D., Hulsman, P., Poyatos, R., & Miralles, D. G. (2022). A deep learning-based hybrid model of global terrestrial evaporation. *Nature Communications, 13*(1), 1912. https://doi.org/10.1038/s41467-022-29543-7

Kuzyakov, Y., & Razavi, B. S. (2019). Rhizosphere size and shape: Temporal dynamics and spatial stationarity. *Soil Biology and Biochemistry, 135*, 343-360.
https://www.sciencedirect.com/science/article/pii/S0038071719301452

Li, B., Rodell, M., & Famiglietti, J. S. (2015). Groundwater variability across temporal and spatial scales in the central and northeastern U.S. *Journal of Hydrology, 525*, 769-780. https://www.sciencedirect.com/science/article/pii/S0022169415002929

Maxwell, R. M., & Condon, L. E. (2016). Connections between groundwater flow and transpiration partitioning. *Science, 353*(6297), 377-380.
https://www.science.org/doi/abs/10.1126/science.aaf7891

McCabe, G. J., & Markstrom, S. L. (2007). *A monthly water-balance model driven by a graphical user interface* (Vol. 1088): US Geological Survey Reston, VA, USA.

McCormick, E. L., Dralle, D. N., Hahm, W. J., Tune, A. K., Schmidt, L. M., Chadwick, K. D., & Rempe, D. M. (2021). Widespread woody plant use of water stored in bedrock. *Nature, 597*(7875), 225-229. https://doi.org/10.1038/s41586-021-03761-3

Miguez-Macho, G., & Fan, Y. (2021). Spatiotemporal origin of soil water taken up by vegetation. *Nature, 598*(7882), 624-628. https://doi.org/10.1038/s41586-021-03958-6

Naumburg, E., Mata-gonzalez, R., Hunter, R. G., McLendon, T., & Martin, D. W. (2005). Phreatophytic Vegetation and Groundwater Fluctuations: A Review of Current Research and Application of Ecosystem Response Modeling with an Emphasis on Great Basin Vegetation. *Environmental Management, 35*(6), 726-740. https://doi.org/10.1007/s00267-004-0194-7

Orellana, F., Verma, P., Loheide II, S. P., & Daly, E. (2012). Monitoring and modeling water-vegetation interactions in groundwater-dependent ecosystems. *Reviews of Geophysics, 50*(3). https://agupubs.onlinelibrary.wiley.com/doi/abs/10.1029/2011RG000383

Ponce-Campos, G. E., Moran, M. S., Huete, A., Zhang, Y., Bresloff, C., Huxman, T. E., et al. (2013). Ecosystem resilience despite large-scale altered hydroclimatic conditions. *Nature, 494*(7437), 349-352. https://doi.org/10.1038/nature11836

Reager, J. T., Thomas, B. F., & Famiglietti, J. S. (2014). River basin flood potential inferred using GRACE gravity observations at several months lead time. *Nature Geoscience, 7*(8), 588-592. https://doi.org/10.1038/ngeo2203

Rohde, M. M., Albano, C. M., Huggins, X., Klausmeyer, K. R., Morton, C., Sharman, A., et al. (2024). Groundwater-dependent ecosystem map exposes global dryland protection needs. *Nature, 632*(8023), 101-107. https://doi.org/10.1038/s41586-024-07702-8

Schlemmer, L., Schär, C., Lüthi, D., & Strebel, L. (2018). A Groundwater and Runoff Formulation for Weather and Climate Models. *Journal of Advances in Modeling Earth Systems, 10*(8), 1809-1832. https://agupubs.onlinelibrary.wiley.com/doi/abs/10.1029/2017MS001260

Singh, C., Wang-Erlandsson, L., Fetzer, I., Rockström, J., & van der Ent, R. (2020). Rootzone storage capacity reveals drought coping strategies along rainforest-savanna transitions. *Environmental Research Letters, 15*(12), 124021. https://dx.doi.org/10.1088/1748-9326/abc377

Stocker, B. D., Tumber-Dávila, S. J., Konings, A. G., Anderson, M. C., Hain, C., & Jackson, R. B. (2023). Global patterns of water storage in the rooting zones of vegetation. *Nature Geoscience*. https://doi.org/10.1038/s41561-023-01125-2

Thompson, S. E., Harman, C. J., Konings, A. G., Sivapalan, M., Neal, A., & Troch, P. A. (2011). Comparative hydrology across AmeriFlux sites: The variable roles of climate, vegetation, and groundwater. *Water Resources Research, 47*(10). https://agupubs.onlinelibrary.wiley.com/doi/abs/10.1029/2010WR009797

Wang, T., Wu, Z., Wang, P., Wu, T., Zhang, Y., Yin, J., et al. (2023). Plant-groundwater interactions in drylands: A review of current research and future perspectives. *Agricultural and Forest Meteorology, 341*, 109636. https://www.sciencedirect.com/science/article/pii/S0168192323003271

---

## Author Comment (AC3)

In the following responses, reviewers' comments are reproduced in their entirety in black, and the authors' responses are noted in blue.

**Reviewer 3**

Title: Substantial root-zone water storage capacity observed by GRACE and GRACE/FO

Author(s): Meng Zhao et al.

MS No.: egusphere-2024-1939

Reviewer: The manuscript derives "root water storage capacity" (Sr) from GRACE and GRACE-FO observations of terrestrial water storage (TWS), along with uncertainty estimates. The GRACE-based Sr estimates are compared to Sr estimates derived (i) from soil parameters (soil depth and soil water holding capacity) and (ii) water balance estimates (using precipitation and evapotranspiration [ET] observations). The authors find that the GRACE-based Sr estimates are 50% larger than those derived from water balance estimates and 380% than those derived from soil parameters. The different Sr estimates are further used to parameterize a USGS "bucket model", with TWS and ET output from the model validated against GRACE TWS observations and ET estimates from a water balance approach. Finally, the authors find that their GRACE-based Sr estimates correlate "realistically" with vegetation productivity data.

The authors address a clear need for accurate estimates of root zone water storage capacity, a topic of interest to HESS readers. However, the findings of the manuscript are not supported with independent observations and are largely circular. It is no surprise that the GRACE-based Sr estimates have a relatively lower error against GRACE-based TWS observations. Specifically, the GRACE-based Sr estimates essentially reflect the range of the GRACE TWS observations, and the NSE metrics primarily measures skill in terms of the mean-square error (MSE). Additionally, it remains unclear to me how the authors remove the groundwater signal from the TWS observations. I recommend that the manuscript be rejected.

Response: We appreciate the reviewer's feedback. In the revised manuscript, we will perform a new validation effort that employs independent datasets that do not have GRACE/FO inputs. This validation approach will be as robust as those used in prior studies. Below, we summarize key validation methods used in similar studies, clarify the rationale for the revised validation approach that we will adopt, and outline how we will present the strengths and limitations of this new validation effort.

1. *Challenges in validating $S_r$ and methods used in previous studies:* Validating large-scale $S_r$ remains inherently difficult because direct measurement of $S_r$ is challenging. Previous studies have primarily employed two indirect validation methods:

    a. Rooting-depth comparison: Stocker et al. (2023) converted their deficit-based $S_r$ estimates (~5 km resolution) into rooting-depths using soil texture and water-holding capacity parameters, and then compared them to field rootingdepth measurements aggregated at biome levels to mitigate the scale mismatch. However, this approach is not suitable for our study. Resolved at a much coarser resolution (~300 km), GRACE/FO-derived $S_r$ samples multiple biome types within a single observational footprint, making biome-level aggregation less meaningful. Additionally, the rooting-depth validation method overlooks groundwater contributions to $S_r$, which Stocker et al. (2023) found to be significant in over half of their measurement sites. This omission will likely become more critical at the spatial scale of GRACE/FO, which averages larger areas and includes more diverse biome types. These factors make the rooting-depth comparison unsuitable for evaluating GRACE/FO-derived $S_r$.

    b. Implementation in a hydrological model: Wang-Erlandsson et al. (2016) used deficit-based $S_r$ estimates in a simple hydrological model and assessed improvements in simulating hydrologic time series. While this approach better aligns with the spatial scale of GRACE/FO, it faces challenges, too. One is the limited availability of high-quality global hydrologic data, which can lead to a circular use of the same data for both $S_r$ estimation and model evaluation, as Wang-Erlandsson et al. (2016) did with satellite-based ET data. This reduces the independence of the validation process. Additionally, the mechanistic linkage between $S_r$ and commonly used hydrological indicators (e.g., ET and streamflow) is complex – pinpointing decisive indicators that are strongly sensitive to $S_r$ is an important research topic yet to be addressed in the literature. Resolving such a complex relationship can be further complicated by the structural errors or uncertainties in other parameters adopted in the model. Together, these challenges can obscure the true impact of accurate $S_r$ parameterization on ecohydrology. For example, in our study, streamflow simulated by the USGS model is mainly driven by precipitation and shows little sensitivity to $S_r$, similar to what was described in the open peer review file of Wang-Erlandsson et al. (2016), which also did not use streamflow measurements for model evaluation.

2. *Revised validation approach and its rationale*: We will use the latest version (v4.1) of the Global Land Evaporation Amsterdam Model (GLEAM) ET dataset (https://www.gleam.eu/) to validate our model results. The GLEAM ET addresses key shortcomings present in other gridded ET products. For example, it combines hybrid learning from eddy-covariance and sap flow to capture vegetation response to drought more accurately (Koppa et al., 2022), and it explicitly accounts for plant access to groundwater (Hulsman et al., 2023). Importantly, the GLEAM ET is independent of GRACE/FO and, therefore, allows robust validation that is free from circularity. To mitigate the impact of possible biases embedded in GLEAM ET, forcing data, and those caused by model uncertainty (as the USGS model is uncalibrated), we will use standardized ET anomalies (i.e., Z-scores) as the target of validation and focus on assessing whether $S_r$ improves the temporal dynamics of ET simulations (i.e., seasonal and interannual variations) rather than the absolute values of ET.

3. *Strengths and limitations of the proposed new validation efforts*: The key strength of our revised validation approach lies in its use of an independent dataset (GLEAM

ET), which addresses the potential circularity of our current validation efforts. We will also examine and discuss the following limitations in the revised manuscript.

a. Challenges in detecting $S_r$ influence: Given the uncertainties of modeling $S_r$'s role in ET dynamics, the improvements in $S_r$ may be challenging to detect, particularly when using large-scale models that rely heavily on precipitation-driven processes. We will examine if and to what extent this can be mitigated by using standardized ET anomalies (Z-scores) as the validation target.

b. Focus on temporal dynamics over absolute values: The proposed use of standardized ET anomalies (Z-scores) shifts the focus from absolute ET values to temporal dynamics (seasonal and interannual variations). While this helps mitigate the impact of data biases, it may also limit the scope of the validation to detecting only temporal variations and not necessarily capturing the full range of hydrological dynamics influenced by $S_r$.

Despite these limitations, the revised validation effort will represent a substantial improvement over Wang-Erlandsson et al. (2016) by using independent, high-quality ET data and focusing on the temporal dynamics of ET.

We will also clarify the definition of root zone storage capacity ($S_r$), acknowledging the inclusion of natural groundwater fluctuations to meet plant water demands, supported by recent studies and field evidence, including our comparison dataset $S_r^{accum}$ from Stocker et al. (2023). Although this broadens the traditional definition of the root zone, it helps delineate the true amount of water available to plants and is consistent with evolving research on groundwater-vegetation interactions.

Major comments:
- Reviewer: The validation approach is circular (contrary to the statement in Lines 137-140). The GRACE-based Sr estimates reflect, by construction, approximately the dynamic range of the validating GRACE TWS observations (as shown in Figure 1). The surface meteorological forcing inputs to the USGS model are the same for all three simulations, and the only difference between the USGS model configurations is in the Sr parameters. The simulated TWS and ET will therefore have very similar *standardized* anomalies (Z-scores), and the key determinant of the NSE metric will be whether the dynamic range of the simulated TWS anomalies matches that of the verifying observations. The latter were used to determine the GRACE-based Sr, thereby essentially guaranteeing a lower MSE and higher NSE for the simulation with the GRACE-based Sr relative to the other simulations. (As an aside, Line 226 refers to "performance in simulating TWS temporal dynamics". This is a bit of an overstatement given the fact that the experiment design primarily measures how well the estimated Sr reflects the dynamic range of the TWS observations. "Temporal dynamics" suggests skill differences in seasonal and interannual variations, which are not explicitly examined and which are likely to be small, given the experiment setup.)
  Response: You raised a good point here. To minimize the influence of dynamic range on the NSE metric, we reanalyzed our results using standardized anomalies (i.e., Z-scores) for both simulated and GRACE/FO-observed TWS time

series. By using Z-scores, we standardized the dynamic range while preserving temporal dynamics, including seasonal and interannual variations. Contrary to the reviewer's assumption, our analysis shows that, even after standardizing the anomalies, the TWS simulations with different $S_r$ parameterizations exhibit distinct patterns.

For example, Fig. RC3_1a compares the Z-scores of TWS from GRACE/FO and three simulations (*HydroModel*($S_r^{GRACE/FO}$), *HydroModel*($S_r^{RD \times WHC}$), and *HydroModel*($S_r^{accum}$)) for the mascon location in Figure 1 of the original submission. The NSE values for the Z-scores time series indicate that $S_r^{GRACE/FO}$ outperforms $S_r^{RD \times WHC}$ and $S_r^{accum}$ in capturing TWS temporal dynamics (Fig. RC3_1b-d). This improvement is widespread (Fig. RC3_2) and overlaps with those based on the original time series (Figure 5 of the original text). Notably, this enhancement extends into many subtropical and Southern Hemisphere regions, where the USGS model struggles to simulate the dynamic range of GRACE/FO TWS.

To address the circularity concern, we will no longer use GRACE/FO TWS as a validation dataset. Instead, we will validate the model using GLEAM ET and evaluate it using standardized ET anomalies (i.e., Z-scores). GLEAM ET is independent of GRACE/FO and, therefore, allows robust validation and avoid the circularity concern. The Z-score approach also allows us to assess the model's ability to capture seasonal and interannual variations without undue influence from potential biases embedded in GLEAM ET, forcing data, and those caused by the uncalibrated nature of the USGS model.

[Figure]

**Figure RC3_1**. A comparison of model predictive skills for TWS z-scores. (a) Z-score time series comparison between GRACE/FO TWS and model simulations. (b)-(d) Scatterplots of

GRACE/FO TWS z-scores and simulated TWS z-scores from *HydroModel(S$_r$$^{GRACE/FO}$)*, *HydroModel(S$_r$$^{RD×WHC}$)*, *HydroModel(S$_r$$^{accum}$)*, respectively.

[Figure]

**Figure RC3_2**. Predictive skill differences for TWS z-scores. (a) The NSE difference between *HydroModel(S$_r$$^{GRACE/FO}$)* and *HydroModel(S$_r$$^{RD×WHC}$)*. (b) The NSE difference between *HydroModel(S$_r$$^{GRACE/FO}$)* and *HydroModel(S$_r$$^{accum}$)*. The gray colors indicate areas where all models fail to achieve a positive NSE value.

- Reviewer: The ET estimates used to validate the USGS model simulations are based on water balance estimates derived from precipitation and water storage change datasets, which is similarly circular when it comes to validating the model output from the simulations that use Sr estimates based on GRACE observations or water balance estimates.
  Response: In our revised manuscript, we will use GLEAM ET, which is independent of GRACE/FO and addresses key shortcomings in other gridded ET products.

- Reviewer: The definition of Sr as "root zone storage capacity" seems inconsistent the derivation from GRACE TWS observations. The authors explain how they remove the snow signal and anthropogenic groundwater signals from the TWS observations when they derive the GRACE-based Sr estimates. However, it remains unclear how natural groundwater fluctuations are handled. TWS observations include natural variations in groundwater levels that are not related

to water storage in what would usually be considered the "root zone" (e.g., in grasslands).   Perhaps it is intentional that such fluctuations are included, but then the derived parameter is then no longer a "root zone storage capacity" in the sense that the control volume is no longer what is commonly understood to be the "root zone".

Response: Thank you for this comment. Natural groundwater variability is indeed embedded in our calculation of root-zone water storage capacity ($S_r$), and we provide clarification below. As Reviewer#1 correctly pointed out, root-accessible water does not require roots to physically occupy the entire storage domain. Processes such as the capillary rise can move deep water upward to the root zone for vegetation transpiration, especially during dry seasons and droughts. Many studies have shown that natural groundwater variability (such as its seasonal variation) strongly correlates with the net effect of precipitation and ET (e.g., Li et al., 2015).

Including groundwater in the calculation of $S_r$ extends the traditional definition of the "root zone," beyond the soil layer by recognizing the fact that the root zone is dynamic and can access deep groundwater and bedrock moisture during prolonged droughts and high transpiration demand (Gao et al., 2024). Several recent studies (McCormick et al., 2021; Singh et al., 2020; Stocker et al., 2023) have also included groundwater in their definitions of $S_r$. This inclusion is well-supported by recent studies based on in situ groundwater (Baldocchi et al., 2021; Fan et al., 2017; Li et al., 2015; Thompson et al., 2011), remote sensing observations (Koirala et al., 2017; Rohde et al., 2024), and modeling efforts (Hain et al., 2015; Miguez-Macho & Fan, 2021), all of which showed that groundwater significantly contributes to ET and is accessible to plants, especially during extreme droughts.

In many ecosystems, water stress can stimulate root growth into deep subsurface through the capillary rise effect, with roots extending to the capillary fringe and the water table, as observed in both field and laboratory studies (Fan et al., 2017; Kuzyakov & Razavi, 2019; Naumburg et al., 2005; Orellana et al., 2012). Although individual shallow-rooted plants (e.g., grassland sites) may not directly tap into groundwater, the large spatial scale of GRACE/FO data likely captures water uptake across diverse vegetation types. This blending makes it likely that vegetation types not typically associated with groundwater use may still access it indirectly, such as through hydraulic redistribution by neighboring deeper-rooted plants (e.g., Espeleta et al., 2004; Orellana et al., 2012). Indeed, satellite observations have revealed widespread plant-groundwater interactions at large spatial scales (Koirala et al., 2017), even in dryland regions dominated by grasslands (Rohde et al., 2024; Wang et al., 2023).

Neglecting groundwater in root zone storage capacity can lead to underestimation of land and air interactions (Dong et al., 2022; Maxwell & Condon, 2016; Schlemmer et al., 2018), affect accurate simulation of runoff (Hahm et al., 2019), and misrepresent vegetation resilience to droughts and heat waves (Esteban et al., 2021; Jiménez-Rodríguez et al., 2022).

Overall, our $S_r^{GRACE/FO}$ definition aligns with our comparison dataset $S_r^{accum}$ from Stocker et al. (2023) and helps explain why the traditional rooting depth approach ($S_r^{RD \times WHC}$), which does not include groundwater, yields lower values than $S_r^{GRACE/FO}$ and $S_r^{accum}$. This expanded definition is consistent with emerging research on groundwater-vegetation interactions. We will add these discussions to the revised manuscript.

- Reviewer: It is highly concerning that no model attains positive NSE values for 40% of the global *vegetated* domain (Lines 216-217). This area includes most of the subtropics and Southern Hemisphere! If the model is so poor that for nearly half of the domain of interest a time-invariant constant would be a better estimator, what does it say about the skill of the model in the other half of the domain? And what does it mean for the Sr estimates in nearly half of the domain of interest where NSE is negative for all three model simulations?
  Response: The USGS model, which was run without any local calibration, failed to have positive NSE values in 40% of the vegetated regions, primarily due to its inability to capture the dynamic range of GRACE/FO-observed TWS (Figs. A4 and A5, and the discussion from lines 328 to 347 of the original submission). This underperformance is likely due to uncalibrated parameters and the model's simplified representation of key hydrological processes. However, by applying the Z-score approach — which minimizes the impact of dynamic range mismatch on the NSE metric — Fig. RC3_2 shows that the USGS model effectively captures TWS temporal variations in many subtropical and Southern Hemisphere regions. The area with no positive NSE values was reduced from 40% to 24%, indicating that the USGS model still provides valuable insights into the relative accuracy of the three $S_r$ estimates in most global vegetated regions.

  Although 24% of the domain continues to show negative NSE values, this does not invalidate the $S_r^{GRACE/FO}$ estimates. Rather, it highlights regions where further investigation and refinement are needed. Future work could involve local calibration of model parameters or using more sophisticated hydrological models to improve accuracy in these challenging areas. Despite the negative NSE values, the $S_r^{GRACE/FO}$ estimate remains informative, offering valuable insights into water storage dynamics when interpreted within the context of known model limitations.

  Given the discussions above, while the Z-score-based GRACE/FO TWS results are informative, we will not include them in the revision. Instead, we will use the GLEAM ET dataset for model validation to ensure our validation is independent of GRACE/FO and free from circularity.

Minor comments:
1. Reviewer: The heading of section 3 should probably be "Results"
   Response: We will use "Results" in our revised manuscript.

2. Reviewer: The caption of Figure 3 does not clearly state the base for the "percentage changes". This can only be understood from the text.
Response: We will change the caption to "(a) and (b) are the consumption percentages **of $S_r^{GRACE/FO}$** during the second and third-largest TWS drawdowns."

3. Reviewer: Line 208: Be more specific about the "drier climates and lower-biomass regions"
Response: We will specify these regions in our revised manuscript.

**References**

Baldocchi, D., Ma, S., & Verfaillie, J. (2021). On the inter- and intra-annual variability of ecosystem evapotranspiration and water use efficiency of an oak savanna and annual grassland subjected to booms and busts in rainfall. *Global Change Biology, 27*(2), 359-375. https://onlinelibrary.wiley.com/doi/abs/10.1111/gcb.15414

Dong, J., Lei, F., & Crow, W. T. (2022). Land transpiration-evaporation partitioning errors responsible for modeled summertime warm bias in the central United States. *Nature Communications, 13*(1), 336. https://doi.org/10.1038/s41467-021-27938-6

Espeleta, J. F., West, J. B., & Donovan, L. A. (2004). Species-specific patterns of hydraulic lift in co-occurring adult trees and grasses in a sandhill community. *Oecologia, 138*(3), 341-349. https://doi.org/10.1007/s00442-003-1460-8

Esteban, E. J. L., Castilho, C. V., Melgaço, K. L., & Costa, F. R. C. (2021). The other side of droughts: wet extremes and topography as buffers of negative drought effects in an Amazonian forest. *New Phytologist, 229*(4), 1995-2006. https://nph.onlinelibrary.wiley.com/doi/abs/10.1111/nph.17005

Fan, Y., Miguez-Macho, G., Jobbágy, E. G., Jackson, R. B., & Otero-Casal, C. (2017). Hydrologic regulation of plant rooting depth. *Proceedings of the National Academy of Sciences, 114*(40), 10572-10577. https://www.pnas.org/content/pnas/114/40/10572.full.pdf

Gao, H., Hrachowitz, M., Wang-Erlandsson, L., Fenicia, F., Xi, Q., Xia, J., et al. (2024). Root zone in the Earth system. *EGUsphere, 2024*, 1-30. https://egusphere.copernicus.org/preprints/2024/egusphere-2024-332/

Hahm, W. J., Dralle, D. N., Rempe, D. M., Bryk, A. B., Thompson, S. E., Dawson, T. E., & Dietrich, W. E. (2019). Low Subsurface Water Storage Capacity Relative to Annual Rainfall Decouples Mediterranean Plant Productivity and Water Use From Rainfall Variability. *Geophysical Research Letters, 46*(12), 6544-6553. https://agupubs.onlinelibrary.wiley.com/doi/abs/10.1029/2019GL083294

Hain, C. R., Crow, W. T., Anderson, M. C., & Yilmaz, M. T. (2015). Diagnosing Neglected Soil Moisture Source–Sink Processes via a Thermal Infrared–Based Two-Source Energy Balance Model. *Journal of Hydrometeorology, 16*(3), 1070-1086. https://journals.ametsoc.org/view/journals/hydr/16/3/jhm-d-14-0017_1.xml

Hulsman, P., Keune, J., Koppa, A., Schellekens, J., & Miralles, D. G. (2023). Incorporating Plant Access to Groundwater in Existing Global, Satellite-Based Evaporation Estimates. *Water Resources Research, 59*(8), e2022WR033731. https://agupubs.onlinelibrary.wiley.com/doi/abs/10.1029/2022WR033731

Jiménez-Rodríguez, C. D., Sulis, M., & Schymanski, S. (2022). Exploring the role of bedrock representation on plant transpiration response during dry periods at four forested sites in Europe. *Biogeosciences, 19*(14), 3395-3423.

Koirala, S., Jung, M., Reichstein, M., de Graaf, I. E. M., Camps-Valls, G., Ichii, K., et al. (2017). Global distribution of groundwater-vegetation spatial covariation. *Geophysical Research Letters, 44*(9), 4134-4142. https://agupubs.onlinelibrary.wiley.com/doi/abs/10.1002/2017GL072885

Koppa, A., Rains, D., Hulsman, P., Poyatos, R., & Miralles, D. G. (2022). A deep learning-based hybrid model of global terrestrial evaporation. *Nature Communications, 13*(1), 1912. https://doi.org/10.1038/s41467-022-29543-7

Kuzyakov, Y., & Razavi, B. S. (2019). Rhizosphere size and shape: Temporal dynamics and spatial stationarity. *Soil Biology and Biochemistry, 135*, 343-360. https://www.sciencedirect.com/science/article/pii/S0038071719301452

Li, B., Rodell, M., & Famiglietti, J. S. (2015). Groundwater variability across temporal and spatial scales in the central and northeastern U.S. *Journal of Hydrology, 525*, 769-780. https://www.sciencedirect.com/science/article/pii/S0022169415002929

Maxwell, R. M., & Condon, L. E. (2016). Connections between groundwater flow and transpiration partitioning. *Science, 353*(6297), 377-380. https://www.science.org/doi/abs/10.1126/science.aaf7891

McCormick, E. L., Dralle, D. N., Hahm, W. J., Tune, A. K., Schmidt, L. M., Chadwick, K. D., & Rempe, D. M. (2021). Widespread woody plant use of water stored in bedrock. *Nature, 597*(7875), 225-229. https://doi.org/10.1038/s41586-021-03761-3

Miguez-Macho, G., & Fan, Y. (2021). Spatiotemporal origin of soil water taken up by vegetation. *Nature, 598*(7882), 624-628. https://doi.org/10.1038/s41586-021-03958-6

Naumburg, E., Mata-gonzalez, R., Hunter, R. G., McLendon, T., & Martin, D. W. (2005). Phreatophytic Vegetation and Groundwater Fluctuations: A Review of Current Research and Application of Ecosystem Response Modeling with an Emphasis on Great Basin Vegetation. *Environmental Management, 35*(6), 726-740. https://doi.org/10.1007/s00267-004-0194-7

Orellana, F., Verma, P., Loheide II, S. P., & Daly, E. (2012). Monitoring and modeling water-vegetation interactions in groundwater-dependent ecosystems. *Reviews of Geophysics, 50*(3). https://agupubs.onlinelibrary.wiley.com/doi/abs/10.1029/2011RG000383

Rohde, M. M., Albano, C. M., Huggins, X., Klausmeyer, K. R., Morton, C., Sharman, A., et al. (2024). Groundwater-dependent ecosystem map exposes global dryland protection needs. *Nature, 632*(8023), 101-107. https://doi.org/10.1038/s41586-024-07702-8

Schlemmer, L., Schär, C., Lüthi, D., & Strebel, L. (2018). A Groundwater and Runoff Formulation for Weather and Climate Models. *Journal of Advances in Modeling*

*Earth Systems, 10*(8), 1809-1832.
https://agupubs.onlinelibrary.wiley.com/doi/abs/10.1029/2017MS001260

Singh, C., Wang-Erlandsson, L., Fetzer, I., Rockström, J., & van der Ent, R. (2020). Rootzone storage capacity reveals drought coping strategies along rainforest-savanna transitions. *Environmental Research Letters, 15*(12), 124021. https://dx.doi.org/10.1088/1748-9326/abc377

Stocker, B. D., Tumber-Dávila, S. J., Konings, A. G., Anderson, M. C., Hain, C., & Jackson, R. B. (2023). Global patterns of water storage in the rooting zones of vegetation. *Nature Geoscience*. https://doi.org/10.1038/s41561-023-01125-2

Thompson, S. E., Harman, C. J., Konings, A. G., Sivapalan, M., Neal, A., & Troch, P. A. (2011). Comparative hydrology across AmeriFlux sites: The variable roles of climate, vegetation, and groundwater. *Water Resources Research, 47*(10). https://agupubs.onlinelibrary.wiley.com/doi/abs/10.1029/2010WR009797

Wang, T., Wu, Z., Wang, P., Wu, T., Zhang, Y., Yin, J., et al. (2023). Plant-groundwater interactions in drylands: A review of current research and future perspectives. *Agricultural and Forest Meteorology, 341*, 109636. https://www.sciencedirect.com/science/article/pii/S0168192323003271

Wang-Erlandsson, L., Bastiaanssen, W. G. M., Gao, H., Jägermeyr, J., Senay, G. B., van Dijk, A. I. J. M., et al. (2016). Global root zone storage capacity from satellite-based evaporation. *Hydrol. Earth Syst. Sci., 20*(4), 1459-1481. https://hess.copernicus.org/articles/20/1459/2016/

---

## Author Response (AR1)

Dear Editor,

Thank you for the opportunity to revise and resubmit our manuscript titled *"Substantial Root-Zone Water Storage Capacity Observed by GRACE and GRACE/FO."* We greatly appreciate the thoughtful feedback provided by you and the reviewers, which has helped us significantly improve the manuscript.

In response to the reviewers' comments, we have made significant revisions:

1. *Circularity in model validation*:

To address concerns regarding the independence of our validation dataset, we replaced GRACE-based TWS and ET products with the latest GLEAM ET dataset (v4.1), which is independent of GRACE/FO and resolves the circularity issue raised by all reviewers. To reduce biases from GLEAM ET, its forcing data, and the uncalibrated USGS model, we validated standardized ET anomalies (Z-scores), focusing on seasonal and interannual dynamics rather than absolute ET values. Our results showed that $S_r^{GRACE/FO}$ improves ET simulations, particularly during droughts (new Figs. 5 and 6). These updates are detailed in the revised text (Section 2.4: lines 132–176, and Section 3.3: lines 215–235). We discussed the strengths and limitations of this new validation approach in the new Section 4.4 (lines 317–335), including the limitations of streamflow-based validation as suggested by Reviewer #2 (discussed from lines 330-333).

2. *Groundwater in the proposed root zone storage capacity*:

We addressed how natural groundwater variability is incorporated into $S_r$ by first refining our Introduction text in lines 35-44 to better highlight the importance of including groundwater support for ET into $S_r$. Additionally, we provided a comprehensive discussion on this topic in the new Section 4.3 (lines 276–316). To further clarify the role of anthropogenic groundwater use, we made a new Fig. A2 and explained the rationale for removing anthropogenic groundwater trends using AQUASTAT data, as suggested by Reviewer #1.

3. *Removal of $S_r$-GPP* analysis:

During revision, we discovered an error in interpreting the Miguez-Macho and Fan (2021) dataset, leading to an incorrect $S_r$ estimate from their data. Acknowledging Reviewer #2's comment that the $S_r$-GPP analysis was not central to our main objectives, we removed this analysis to maintain the focus on $S_r^{GRACE/FO}$ and its validation using independent GLEAM ET data in the revised manuscript.

Overall, our conclusions remain unchanged, but we believe these revisions have significantly enhanced the clarity, robustness, and scientific merits of this manuscript. A detailed point-by-point response to all reviewer comments is included in the response letter. In the following pages, reviewers' comments are reproduced in their entirety in black, and our responses are noted in blue.

We hope you will find our revised manuscript acceptable.

Best regards,
Meng Zhao
On behalf of all co-authors

**Reviewer 1**

Reviewer: Review of "Substantial root-zone water storage capacity observed by GRACE and GRACE/FO" by Zhao et al. . This paper describes the use of TWS estimates from the GRACE satellite project to estimate multi-year water storage changes. Negative changes are used to estimate a lower-bound on root-zone water storage ($S_r$). The estimates are compared to two alternative ($S_r$) methodologies, and all three $S_r$ estimates are used to parameterize a hydrologic model. The main result is that the authors' $S_r$ is significantly larger than the previously described $S_r$ estimates.

Comments

General

I found the authors' use of GRACE data to be novel, and the results interesting. The paper is well-written and generally clear.
Response: Thanks for the positive feedback.

As with other GRACE studies, the spatial resolution of the data is relatively coarse, so I suggest adding some discussion of how these results might be applied in the operational configuration of land models, which would typically have finer spatial resolution.
Response: Thanks for your comment and suggestion. In our revised manuscript, we discussed two ways to use our $S_r$ estimates and methodology for land models. First, $S_r^{GRACE/FO}$ can be used for evaluating model default $S_r$ parameterization at the coarse-spatial scale of GRACE/FO data in conjunction with other analyses. For instance, if a model simulates low ET during droughts in a region where the $S_r$ value is also low compared to $S_r^{GRACE/FO}$, the default value may be increased based on $S_r^{GRACE/FO}$ even if the model's resolution is much higher than that of $S_r^{GRACE/FO}$. Second, we discussed approaches for developing finer-scale GRACE-based $S_r$ products, such as using downscaled TWS products developed through machine learning and data assimilation techniques (Gou & Soja, 2024; Li et al., 2019). In our revision, we added a new discussion section from lines 336 to 344, which are reproduced below:

> *"4.5 Implications for high-resolution land surface models*
> *Despite the coarse resolution of GRACE/FO observations, $S_r^{GRACE/FO}$ and our proposed approach remain valuable for improving the operational configuration of higher-resolution land models. First, $S_r^{GRACE/FO}$ can be used to evaluate and refine default $S_r$ parameterizations within models once aggregated to coarse scale of GRACE/FO data, in conjunction with other diagnostic analyses. For instance, if a model underestimates ET during droughts in a region where its $S_r$ value is significantly lower than $S_r^{GRACE/FO}$, the default $S_r$ value may be increased based on $S_r^{GRACE/FO}$ even if the model's resolution is much higher than that of $S_r^{GRACE/FO}$. Second, in the future, our methodology can be extended to downscaled GRACE/FO products, leveraging techniques such as data*

*assimilation systems or artificial intelligence to improve the spatial resolution of $S_r^{GRACE/FO}$ (Li et al., 2019; Gou and Soja, 2024). "*

Reviewer: Abstract
The maximum water held would be the difference from saturation to wilting point. But saturated conditions are unlikely to occur at these spatial scales in many regions.
Response: Our root zone storage capacity includes water uplifted from groundwater, and thus, it is not limited by the wilting point and saturation. In the revision, we rephrased it to "the maximum water volume available for vegetation uptake" in line 11.

Reviewer: 1st sentence defines Sr, and the next sentence discusses simulations. Perhaps add a sentence indicating how Sr is used in a modeling context after the 1st sentence to provide context.
Response: Thanks. We added a sentence to bridge the gap between the definition of $S_r$ and its importance in simulations in lines 13-14: *"In land models, $S_r$ serves as a critical parameter to simulate water availability for vegetation and its impact on processes like transpiration and soil moisture dynamics."*

Reviewer: Line 15: to be clear, GRACE measures gravity and TWS is inferred from that, so the use of the word 'direct' can be problematic. There are other geophysical processes that affect time-varying gravity.
Response: We removed 'direct.'

Reviewer: Line 20: what does 'correlates realistically' mean? Can you use a more specific or quantitative description?
Response: We removed this sentence as part of our decision to exclude GPP-related results in the revision. This change ensures the study remains focused on the calculation of $S_r^{GRACE/FO}$ and its comparison with other datasets and validation using the USGS model.

Reviewer: Introduction
Line 26: 'plants can store during wet periods' should be 'plants can access'? i.e. plants aren't storing the water, the soil is storing water.
Response: We revised the sentence to *"the more water root zone can store during wet periods for use in droughts"* in line 28.

Reviewer: Line 37: why would it overlook rock moisture and groundwater? This sentence implies a different reason besides uncertainties in rooting depths or hydraulic properties, which are mentioned previously.
Response: Thank you for this comment. The reason water stored in weathered rocks and groundwater is often overlooked is that most approaches typically set rooting depth shorter than simulated soil thickness and assume that roots do not extend into deeper unsaturated zones. However, recent studies have shown that this assumption is not always accurate (Rempe and Dietrich, 2018; Fan et al., 2017). In fact, in many ecosystems, plant roots can penetrate beyond the shallow soil layer into weathered

bedrocks to access deep water storage, including groundwater, especially during dry seasons and droughts (McCormick et al., 2021; Maxwell and Condon, 2016).

We clarified this and highlighted the importance of rock moisture and groundwater in our definition of root-zone storage capacity from lines 35 to 44 in the revised manuscript (new text in bold):

> "The $S_r$ is typically calculated as the integration of plant rooting depth and soil texture-dependent water-holding capacity (Seneviratne et al., 2010; Vereecken et al., 2022; Speich et al., 2018; Federer et al., 2003). However, this approach (hereafter referred to as the rooting depth-based estimation) suffers from uncertainties associated with plant rooting depth and substrate hydraulic properties, particularly at depth, both of which undermine the accuracy of the calculated $S_r$ (Vereecken et al., 2022; Novick et al., 2022). **Moreover, this approach assumes a static root zone confined to the near surface unsaturated soil layer. However, recent studies have shown that this assumption is not always accurate. In many ecosystems, plant roots can penetrate beyond the shallow soil layer into weathered bedrock, accessing rock moisture and tapping into groundwater, especially during prolonged dry periods** (Li et al., 2015; Hahm et al., 2020; McCormick et al., 2021; Rempe and Dietrich, 2018; Maxwell and Condon, 2016; Fan et al., 2017; Baldocchi et al., 2021). **Thus, the rooting depth-based estimation may significantly underestimate $S_r$.**"

Reviewer: Line 49: again, the word 'direct' I find problematic. If you wish to use this word, perhaps add a sentence explaining its use.
Response: We removed 'direct.'

Reviewer: Methods
Line 61: clearly, 'root-zone' implies vegetated areas, but what might one learn from this method in more arid regions?
Response: Thank you for your question. In all regions, soil evaporation contributes to our estimate, although we expect its influence to be minor relative to that of transpiration in more vegetated regions. This contribution is likely to be greater in more arid regions, but all vegetated regions have some transpiration and root-zone, and we expect these to be fully depleted in the largest drydown, thereby contributing to our estimate of $S_r$. In more arid regions such as deserts, our approach may capture moisture storage capacity for bare soil evaporation. We discussed this point in the revision from line 106 to 109:

> "Our method also implicitly includes moisture stored in the topsoil for soil evaporation (Stoy et al., 2019). However, the contribution of soil evaporation to ET decreases quickly as TWS draws down (Stocker et al., 2023), and we expect that the magnitude of the largest drawdown will be determined by root-zone depletion magnitude reflected at the end of the drawdown. "

Reviewer: Line 68: typically P, ET, and R refer to fluxes. To be more consistent with other literature, consider using rate or flux units consistently and include a summation symbol in equation 1.
Response: We used flux units and added a summation symbol in equation 1 in the revision.

Reviewer: Line 75: 'consumed' could be changed to 'transpired' or 'returned to the atmosphere'
Response: We changed it to 'transpired' in line 80 of the revised manuscript.

Reviewer: Line 82: in areas in which widespread groundwater use is absent, how will this trend removal affect your results? Is it likely to increase or decrease your Sr estimates for such areas? Could you use maps of irrigated area, such as AQUASTAT, to confine this operation to areas where widespread irrigation occurs?
Response: You raised a good point here. In some cases, long-term negative trends in TWS can be associated with precipitation trends in responses to climate change. In those cases, removing long-term linear trends likely leads to underestimation of $S_r$. However, we found that regions showing significant TWS decreasing trends largely coincide with known irrigation areas identified in AQUASTAT data, except in some high Arctic locations (Figs. R1a, b). Thus, our $S_r$ estimates may be underestimated in these high Arctic regions. We added a discussion of this limitation in the revision from lines 252 to 258:

> "*Additionally, our approach to account for groundwater pumping and surface water may overestimate these signals' actual magnitudes and thus likely contribute to underestimating $S_r$. Specifically, we assumed all negative TWS trends to be caused by groundwater withdrawals and removed them from $S_r^{GRACE/FO}$. However, intense groundwater withdrawals are concentrated in specific regions such as northwest India, California's Central Valley, and the North China Plain (Rodell et al., 2009; Feng et al., 2013; Liu et al., 2022). Consequently, we may have removed TWS depletion trends caused by natural variability, as seen in the drought-stricken Southeast Brazil (Rodell et al., 2018). This likely explains why $S_r^{GRACE/FO}$ is lower than $S_r^{accum}$ there (Fig. 3b)."*

The AQUASTAT dataset has its own uncertainties and limitations. For instance, it is based on statistics from 2000-2008 and is particularly uncertain in high-latitude regions (Fig. R1c). Consequently, it may not provide reliable information on groundwater use in some areas of the globe and we chose not to include it as a part of our analysis. Nevertheless, we believe that its match to our trend removals (as shown in Fig. R1) builds confidence in our analysis. We added Fig. R1 to the revised manuscript as Fig. A2 and added a description in lines 87-89 to provide more context for our negative TWS trend removal:

> "*Anthropogenic groundwater use often manifests as a negative long-term trend in the TWS time series (Rodell et al., 2018; Rodell et al., 2009; Feng et al., 2013). For*

*example, regions showing significant TWS decreasing trends largely coincide with well-known groundwater irrigation areas identified in AQUASTAT data (Fig. A2)."*

[Figure]

**Figure R1.** (a) Trends in TWS obtained from GRACE/FO observations from 2002 to 2022. (b) Percentage of area equipped for irrigation that is actually irrigated. (c) Map quality marks assigned to each country for area equipped for irrigation in (b). (b-c) are from the Global Map of Irrigation Areas – version 5.0 by AQUASTAT available at https://firebasestorage.googleapis.com/v0/b/fao-aquastat.appspot.com/o/PDF%2FMAPS%2Fgmia_v5_lowres.pdf?alt=media&token=d098a48f-ab49-4eae-a16e-82a5779f924e

Reviewer: Line 91: how runoff is used here is not clear to me. Is there a budget equation that could be shown? What does 'surface water' encompass; rivers, lakes, reservoirs, ...?

Response: Yes, surface water here encompasses water stored in rivers, lakes, and reservoirs. We added this clarification in lines 99-100 of the revised manuscript.

In GRACE/FO TWS decomposition studies (Bhanja et al., 2016; Getirana et al., 2017; Shamsudduha & Taylor, 2020; Thomas et al., 2017; S. Wang, J. Li, & H. A. Russell, 2023), surface runoff ($Q$) is commonly used as a proxy for surface water storage change ($\Delta SW$), expressed as $\Delta SW = Q$. This approach assumes $Q$ directly contributes to an increase in surface water levels within the drainage network. This approach also assumes that it takes approximately one month for $Q$ to exit the drainage system, aligning with the monthly time step of GRACE/FO data.

In our study, we used total runoff ($R$), which includes both surface runoff ($Q$) and subsurface runoff, as a proxy for $\Delta SW$ (i.e., $\Delta SW = R$), as subsurface runoff which is groundwater discharge to rivers also contributes to surface water storage changes.

We clarified the methodology and assumptions further in the revised manuscript and included the water budget equation ($\Delta SW = R$) for clarity from lines 97 to 102 (new text in bold):

"*Following Wang et al. (2023a), we used total runoff from Ghiggi et al. (2021),* **which includes both surface runoff and subsurface runoff**, *as a proxy for surface water storage change* **(i.e., $\Delta SW = R$)** *and removed it from TWS drawdowns to isolate $\Delta SW$ contributions to the GRACE/FO signal.* **This approach assumes that (1) R directly contributes to an increase in surface water levels within the drainage network, and (2) it takes approximately one month for R to exit the drainage system, aligning with the monthly time step of GRACE/FO data.**"

Reviewer: Line 109: to what extent is Yang 2016 a model-based dataset versus an observational dataset?

Response: Yang et al. (2016) is a fully model-based dataset. It relies on Guswa (2008)'s analytical model that estimates rooting depth, which makes an assumption about root growth based on the carbon gain and cost of any additional roots. While such model-based datasets are valuable for providing comprehensive coverage and insights into complex processes, they do not incorporate direct observational data for validation or correction. We discussed this caveat in the revision from lines 124 to 126:

"*While such model-based datasets are valuable for providing comprehensive coverage and insights into complex processes, they do not incorporate direct observational data for validation or correction.*"

Reviewer: Line 111: how is water holding capacity defined? Field capacity minus wilting point?

Response: The reviewer is correct. Field capacity is defined as the difference between field capacity and permanent wilting point. We added this definition in our revised

manuscript from lines 126 to 128: "*Soil water holding capacity, **defined as the difference between field capacity and permanent wilting point,** is calculated based on …*"

Reviewer: Line 132: why is this an approximation?  Are there other modeled water storage components in HydroModel that were ignored?
Response: There are no other modeled water storage components in the USGS model. To address concerns about the potential circular use of GRACE/FO data, we revised the manuscript to no longer use TWS as the target validation variable. Consequently, we removed the description of modeled water storage components from the revised manuscript to align with this adjustment.

Reviewer: Line 141: 'ET anomalies'
Response: Thank you. We corrected this in the revised manuscript.

Reviewer: Line 146: Does Xiong 2023 use GRACE water storage for their ET estimates?  If so, does that reduce its independence from your results?
Response: Yes, Xiong et al. (2023) used GRACE for their ET estimates. In the revised manuscript, we have replaced the Xiong et al. (2023) ET estimates with the latest version (v4.1) of the Global Land Evaporation Amsterdam Model (GLEAM) ET dataset (Diego G Miralles et al., 2024) to validate our model results. The GLEAM ET is a state-of-the-art dataset, addressing key issues present in other gridded ET products. For example, it combines hybrid learning from eddy-covariance and sap flow to capture vegetation responses to drought more accurately (Koppa et al., 2022), and it explicitly accounts for plant access to groundwater (Hulsman et al., 2023). Importantly, the GLEAM ET is independent of GRACE/FO and, therefore, allows robust validation that is free from circularity.

Overall, our new validation effort using GLEAM ET suggests $S_r^{GRACE/FO}$ improves ET simulations over the other two estimates, particularly during droughts (new Figs. 5 and 6). We described our new validation effort from lines 132 to 176 (new Section 2.4) and presented the new validation results from lines 215 to 235 (new Section 3.3).

Reviewer: Line 169: you say that you compare the two datasets, but you don't explicitly say what your hypothesized relationship between them is, so the justification here seems weak.  In areas that are not water limited, one could imagine that GPP would be high, but a deep root zone is not necessary.  Perhaps expand further on your reasoning in this paragraph.
Response: Thank you for your comment. We agree that the hypothesized relationship between the datasets was insufficiently justified. To maintain the focus of the study on the calculation of $S_r^{GRACE/FO}$ and its validation using the USGS model, we have removed the analysis involving GPP and its associated discussion from the revised manuscript.

Reviewer: Line 194: is this saying that the durations shown in 3c) and 3d) are often larger than that shown in 2b)?

Response: No. The average duration of the first, second, and third-largest TWS drawdowns are 2.8, 1.6, and 1.2 years, respectively. This indicates that the durations in Fig.3 c) and d) are often shorter than those shown in Fig. 2b.

Reviewer: Line 225: Do these patterns correlate with a particular land cover or vegetation type?
Response: We did not find a clear correlation with a particular land cover or vegetation type. This may be due to the large spatial resolution of GRACE/FO data, which represents combined signals from various land cover types within its 3° × 3° footprint. As a result, it is challenging to isolate patterns specific to individual land cover or vegetation types.

Reviewer: Line 271: plot d) is unclear to me. You create an Sr estimate from Miguez-Macho 2021, but then plot it against transpiration instead of GPP; why is this done differently from a) - c)?
Response: Thank you for pointing this out. The reason for this difference is that Miguez-Macho and Fan (2021) only provided transpiration data, not GPP. Upon further review, we identified an error in our interpretation of the Miguez-Macho and Fan (2021) dataset, which led to an inaccurate estimate of $S_r$ from their data. To maintain the focus of the study on the calculation of $S_r^{GRACE/FO}$ and its validation using the USGS model, we have removed the analysis involving GPP and its associated discussion from the revised manuscript.

Reviewer: Figure 8: why are the x- and y-axis ranges different for plots a) - c)? It is harder to compare the scatterplots because of this.
Response: As discussed in the previous response, we removed this figure from the revised manuscript.

Reviewer: Discussion
Line 321: does root-accessible water require that the roots physically occupy the entire storage domain? For example, as soils dry, upward moisture fluxes can occur which might replenish soil moisture deficits near roots. Might this help explain the mismatch between observed rooting depths and the Sr estimates here?
Response: Thank you for this insightful comment. You are correct that roots do not necessarily need to physically occupy the entire storage domain. Processes such as the capillary force can indeed move deep water storage upward to replenish moisture near the roots, especially during dry conditions. Such a mechanism could be the reason for the observed differences between the rooting depth-based estimation and our GRACE/FO-based estimation. We included this discussion in the revised manuscript from lines 283 to 285:

> "*Indeed, root-accessible water does not require roots to physically occupy the entire storage domain. Processes like capillary rise can move deep water upward to the traditional "root zone" for vegetation transpiration, especially during dry seasons and droughts.*"

Reviewer: Line 325: one could also interpret your Sr/WHC as simply the effective soil depth. For land models that do not use an explicit Sr variable, this could indicate that models with a soil depth < 2m (i.e. some of the GLDAS models) are likely incapable of simulating these kinds of drawdowns, which would have implications for studies of groundwater that have used GLDAS to remove the soil moisture component of TWS.

Response: Agreed. We discussed these implications in the revised manuscript from lines 312 to 316 (new text in bold):

> "*These results indicate that the potential for plants to tap into deep water stores is more prevalent than previously understood. **For land models that do not explicitly incorporate $S_r$ as a variable, this suggests that models with a soil depth of less than 2 m (e.g., the Noah model within the Global Land Data Assimilation System (GLDAS)) may be unable to accurately simulate these deeper water drawdowns. Consequently, this limitation could impact studies of groundwater that rely on GLDAS to separate soil moisture from TWS (e.g., Rodell et al., 2009).***"

Reviewer: Line 326: 'tap'

Response: Corrected. Thank you.

Reviewer: Figure A1: how does this result relate to the relationship between magnitude and duration? Does it imply that during the largest drawdowns, there is also the largest 'net precipitation'? That seems counterintuitive.

Response: Thank you for your observation. In the previous version of the manuscript, Figure A1 showed the *cumulative sum* of P – R during the drawdown periods. The largest drawdown mostly corresponds to the longest duration, which results in a higher cumulative sum of P – R. We recognize that this might seem counterintuitive, as it suggests that the largest drawdowns also have the largest 'net precipitation.' To clarify this, we revised the figure to present the average P – R instead of the *cumulative sum*. This adjustment will remove the influence of duration and reflect the mean P – R during drawdown periods.

[Figure]

*Revised Figure A1*. *The average P - R during the largest (a), the second largest (b), the third largest (c), the fourth largest (d), and the fifth largest (e) TWS drawdowns.*

**Reviewer 2**

Reviewer: In this work, the authors use the Gravity Recovery and Climate Experiment (GRACE) and GRACE Follow-on (FO) to estimate root-zone storage capacity (Sr). They find estimates of Sr that are much larger than those using mass-balance approaches and rooting depth parameterizations. I found the work interesting, and the writing was succinct and clear. However, I had a difficult time understanding the assumptions and the implications of these assumptions to evaluate the results. I think the authors need to be much clearer about the implications of their assumptions.

Response: Thanks for your overall positive comment. We have tried to clarify our assumptions and their implications throughout the revised manuscript, as discussed in more detail below.

Main comments:
- Reviewer: The proposed method is quite different from previous work because it directly uses total water storage (TWS) from GRACE. However, GRACE measures a combination of surface water, groundwater, soil moisture, snow and ice. You explain how you remove the streamflow and snow/ice…but how do you remove the effect of groundwater? Are you assuming that groundwater is part of Sr? In some cases, as water table becomes more shallow, conditions become anoxic for plants…wouldn't this decrease Sr? The role of gw in Sr calculations must be better explained and the assumptions clearly laid out.

  Response: Thank you for this comment. Here we outline how groundwater is incorporated into our calculations and clarify the underlying assumptions.

  Our $S_r$ estimate includes groundwater. Specifically, we assume that groundwater is an integral component of $S_r$. As Reviewer 1 correctly pointed out that root-accessible water does not require roots to physically occupy the entire storage domain. Processes like capillary rise can move deep water upward to the traditional "root zone" for vegetation transpiration, especially during dry seasons and droughts. This broadens the traditional definition of the "root zone," which is only limited to the surface unsaturated soil layer, to recognize that plants can access deep water stores, including groundwater and rock moisture, during periods of high water demand. This is consistent with recent studies (e.g., McCormick et al., 2021; Miguez-Macho & Fan, 2021; Stocker et al., 2023) that have similarly included groundwater as part of $S_r$. Overall, our assumption aligns with our comparison dataset $S_r^{accum}$ from Stocker et al. (2023) and helps explain why the traditional rooting depth approach ($S_r^{RD \times WHC}$), which does not include groundwater, yields lower values than $S_r^{GRACE/FO}$ and $S_r^{accum}$.

  To address this point more explicitly in the manuscript, we have refined our Introduction text in lines 35-44 to better highlight the importance of including groundwater support for ET into $S_r$. We also added a new discussion section (Section 4.3) to explicitly describe the role of groundwater in our $S_r$ calculation, the assumptions behind this inclusion, and how it aligns with our comparison datasets from lines 276 – 302 in the revised manuscript.

- Reviewer: The proposed method is also quite different from previous methods in the spatial and temporal scale. You are looking at monthly data at 3x3 degrees. This would include several ecosystems that behave very differently. It also includes multi-year droughts…whereas other calculations would account for periods of deficit (calculated at the daily timescale) with a certain return period. This is a completely different metric…is it really appropriate to compare these?
  Response: We appreciate the reviewer's comment. Indeed, our calculation is fundamentally different from previous approaches. However, we contend that the comparability of the two metrics rests on their shared definition of the physical processes involved. Both $S_r^{accum}$ from Stocker et al. (2023) and $S_r^{GRACE/FO}$ define root zone storage capacity in an identical manner, encompassing groundwater and rock moisture and averaging across diverse ecosystems at large spatial scales.

- Reviewer: I am having a difficult time understanding physically what it means to calculate deltaTWS as the difference between TWS anomalies. Are you assuming that the soil will be at saturation at the beginning of the drawdawn, but will never reach saturation throughout the drawdawn period? Is this an appropriate assumption?
  Response: No, our method does not assume soil saturation at the beginning of the drawdown. In fact, saturation is unlikely to occur at the spatial and temporal scales of $S_r^{GRACE/FO}$. During the drawdown period, deltaTWS represents the water, in equivalent water heights (mm), that an ecosystem has used for ET consumption beyond what is available from effective precipitation (precipitation minus total runoff). This calculation does not require saturation.

- Reviewer: I don't think you should use GRACE to evaluate the performance of HydrModel that includes GRACE information. You state that this is not circular…but it is. Another metric could be streamflow, it would be independent.
  Response: Agreed. In the revised manuscript, we evaluated the model performance with the latest version (v4.1) of the Global Land Evaporation Amsterdam Model (GLEAM) ET dataset (Diego G Miralles et al., 2024). The GLEAM ET is a state-of-the-art dataset, addressing key issues present in other gridded ET products. For example, it combines hybrid learning from eddy-covariance and sap flow to capture vegetation response to drought more accurately (Koppa et al., 2022), and it explicitly accounts for plant access to groundwater (Hulsman et al., 2023). Importantly, the GLEAM ET is independent of GRACE/FO and, therefore, allows validation that is free from circularity.

  We appreciate your suggestion to use streamflow for model evaluation. Unfortunately, streamflow is not the most reliable measure for evaluating the USGS model. First, the USGS model primarily parameterizes streamflow based on precipitation, with subsurface storage contributing only when the storage "bucket" is full (McCabe & Markstrom, 2007). This oversimplified scheme does not adequately represent base flow, which is more directly influenced by water stored in the subsurface including groundwater (Reager et al., 2014). Second, the two key parameters governing streamflow generation – the fraction of

precipitation converted to direct runoff and the fraction of spillover from the storage bucket converted to runoff – are globally uniform and not calibrated to local conditions. This lack of calibration limits the model's capability to capture spatial and temporal variability in streamflow dynamics. Third and more importantly, compared to precipitation, ET, and TWS anomalies, streamflow is the smallest component of the Earth's hydrological cycle. As a result, it is less sensitive to $S_r$ parameterizations. For these reasons, we used ET as an evaluation metric (following Wang-Erlandsson et al. (2016)) in the revised manuscript.

Overall, our new validation effort using GLEAM ET suggests $S_r^{GRACE/FO}$ improves ET simulations over the other two estimates, particularly during droughts (new Figs. 5 and 6). We described our new validation effort from lines 132 to 176 (new Section 2.4) and presented the new validation results from lines 215 to 235 (new Section 3.3).

- Reviewer: The part about linking Sr to vegetation growth was not very convincing. I think you are comparing maximum GPP to the point of saturation…so if I understand correctly what you are showing is that vegetation activity is enhanced when there is enough water. I don't think this argument is necessary for your paper.
  Response: Thank you for your comment. We agree that the argument linking $S_r$ to vegetation growth via maximum GPP could be clearer and, as you suggest, may not be necessary for the core objective of our study. Upon further reflection, we have decided to remove this analysis and its associated discussion to maintain our focus on the calculation of $S_r^{GRACE/FO}$ and its validation using independent GLEAM ET data. This elimination streamlines the manuscript and ensures a sharper focus on the primary contributions of our work.

**Reviewer 3**

Title: Substantial root-zone water storage capacity observed by GRACE and GRACE/FO

Author(s): Meng Zhao et al.

MS No.: egusphere-2024-1939

Reviewer: The manuscript derives "root water storage capacity" (Sr) from GRACE and GRACE-FO observations of terrestrial water storage (TWS), along with uncertainty estimates. The GRACE-based Sr estimates are compared to Sr estimates derived (i) from soil parameters (soil depth and soil water holding capacity) and (ii) water balance estimates (using precipitation and evapotranspiration [ET] observations). The authors find that the GRACE-based Sr estimates are 50% larger than those derived from water balance estimates and 380% than those derived from soil parameters. The different Sr estimates are further used to parameterize a USGS "bucket model", with TWS and ET output from the model validated against GRACE TWS observations and ET estimates from a water balance approach. Finally, the authors find that their GRACE-based Sr estimates correlate "realistically" with vegetation productivity data.

The authors address a clear need for accurate estimates of root zone water storage capacity, a topic of interest to HESS readers. However, the findings of the manuscript are not supported with independent observations and are largely circular. It is no surprise that the GRACE-based Sr estimates have a relatively lower error against GRACE-based TWS observations. Specifically, the GRACE-based Sr estimates essentially reflect the range of the GRACE TWS observations, and the NSE metrics primarily measures skill in terms of the mean-square error (MSE). Additionally, it remains unclear to me how the authors remove the groundwater signal from the TWS observations. I recommend that the manuscript be rejected.
Response: We appreciate the reviewer's feedback. Our detailed responses to your specific comments, which align with the objections raised in your summary paragraph, are provided below.

Major comments:
- Reviewer: The validation approach is circular (contrary to the statement in Lines 137-140). The GRACE-based Sr estimates reflect, by construction, approximately the dynamic range of the validating GRACE TWS observations (as shown in Figure 1). The surface meteorological forcing inputs to the USGS model are the same for all three simulations, and the only difference between the USGS model configurations is in the Sr parameters. The simulated TWS and ET will therefore have very similar *standardized* anomalies (Z-scores), and the key determinant of the NSE metric will be whether the dynamic range of the simulated TWS anomalies matches that of the verifying observations. The latter were used to determine the GRACE-based Sr, thereby essentially guaranteeing a lower MSE and higher NSE for the simulation with the GRACE-based Sr

relative to the other simulations.  (As an aside, Line 226 refers to "performance in simulating TWS temporal dynamics".  This is a bit of an overstatement given the fact that the experiment design primarily measures how well the estimated Sr reflects the dynamic range of the TWS observations.  "Temporal dynamics" suggests skill differences in seasonal and interannual variations, which are not explicitly examined and which are likely to be small, given the experiment setup.)

Response: You raised a good point here. To minimize the influence of dynamic range on the NSE metric, we reanalyzed our results using standardized anomalies (i.e., Z-scores) for both simulated and GRACE/FO-observed TWS time series. By using Z-scores, we standardized the dynamic range while preserving temporal dynamics, including seasonal and interannual variations. Contrary to the reviewer's assumption, our analysis shows that, even after standardizing the anomalies, the TWS simulations with different $S_r$ parameterizations exhibit distinct patterns.

For example, Fig. R2 compares the Z-scores of TWS from GRACE/FO and three simulations (*HydroModel*($S_r^{GRACE/FO}$), *HydroModel*($S_r^{RD \times WHC}$), and *HydroModel*($S_r^{accum}$)) for the mascon location in Figure 1 of the original submission. The NSE values for the Z-scores time series indicate that $S_r^{GRACE/FO}$ outperforms $S_r^{RD \times WHC}$ and $S_r^{accum}$ in capturing TWS temporal dynamics (Fig. R2b-d).  This improvement is widespread (Fig. R3) and overlaps with those based on the original time series (Figure 5 of the original submission). Notably, this enhancement extends into many subtropical and Southern Hemisphere regions, where the USGS model struggles to simulate the dynamic range of GRACE/FO TWS.

To avoid the circularity concern, we no longer use GRACE/FO TWS as a validation dataset. Instead, we validated the model using GLEAM ET version 4.1 (Diego G Miralles et al., 2024) and evaluated it using standardized ET anomalies (i.e., Z-scores). GLEAM ET is independent of GRACE/FO and, therefore avoids the circularity concern. The Z-score approach also allows us to assess the model's ability to capture seasonal and interannual variations without undue influence from potential biases embedded in GLEAM ET, forcing data, and those caused by the uncalibrated nature of the USGS model. Overall, our new validation effort using GLEAM ET suggests $S_r^{GRACE/FO}$ improves ET simulations over the other two estimates across much of the globe, particularly during droughts (new Figs. 5 and 6). We described our new validation effort from lines 132 to 176 (new Section 2.4) and presented the new validation results from lines 215 to 235 (new Section 3.3).

[Figure]

**Figure R2**. A comparison of model predictive skills for TWS z-scores. (a) Z-score time series comparison between GRACE/FO TWS and model simulations. (b)-(d) Scatterplots of GRACE/FO TWS z-scores and simulated TWS z-scores from $HydroModel(S_r^{GRACE/FO})$, $HydroModel(S_r^{RD\times WHC})$, $HydroModel(S_r^{accum})$, respectively.

[Figure]

**Figure R3**. Predictive skill differences for TWS z-scores. (a) The NSE difference between $HydroModel(S_r^{GRACE/FO})$ and $HydroModel(S_r^{RD\times WHC})$. (b) The NSE difference between $HydroModel(S_r^{GRACE/FO})$ and $HydroModel(S_r^{accum})$. The gray colors indicate areas where all models fail to achieve a positive NSE value.

- Reviewer: The ET estimates used to validate the USGS model simulations are based on water balance estimates derived from precipitation and water storage change datasets, which is similarly circular when it comes to validating the model output from the simulations that use Sr estimates based on GRACE observations or water balance estimates.
Response: As discussed above, in our revised manuscript, we used GLEAM ET (v.4.1), which is independent of GRACE/FO and addresses key shortcomings in other gridded ET products.

- Reviewer: The definition of Sr as "root zone storage capacity" seems inconsistent the derivation from GRACE TWS observations. The authors explain how they remove the snow signal and anthropogenic groundwater signals from the TWS observations when they derive the GRACE-based Sr estimates. However, it remains unclear how natural groundwater fluctuations are handled. TWS observations include natural variations in groundwater levels that are not related to water storage in what would usually be considered the "root zone" (e.g., in grasslands). Perhaps it is intentional that such fluctuations are included, but then the derived parameter is then no longer a "root zone storage capacity" in the sense that the control volume is no longer what is commonly understood to be the "root zone".
Response: Thank you for this comment. Our $S_r^{GRACE/FO}$ calculation includes groundwater as an integral component. As Reviewer 1 correctly pointed out that root-accessible water does not require roots to physically occupy the entire storage domain. Processes like capillary rise can move deep water upward to the traditional "root zone" for vegetation transpiration, especially during dry seasons and droughts. In many ecosystems, water stress can stimulate root growth into deep subsurface through the capillary rise effect, with roots extending to the capillary fringe and the water table, as observed in both field and laboratory studies (Fan et al., 2017; Kuzyakov & Razavi, 2019; Naumburg et al., 2005; Orellana et al., 2012). The inclusion of groundwater in the definition of $S_r$ aligns with recent studies (e.g., Fan et al., 2017; McCormick et al., 2021; Miguez-Macho & Fan, 2021; Stocker et al., 2023) that similarly incorporate groundwater into their $S_r$ definitions. Moreover, it is consistent with the comparison dataset $S_r^{accum}$ from Stocker et al. (2023) and explains why the traditional rooting depth approach ($S_r^{RDxWHC}$), which excludes groundwater, yields lower values than both $S_r^{GRACE/FO}$ and $S_r^{accum}$.

We acknowledge that in some regions, especially where shallow-rooted vegetation dominates (e.g., grasslands), groundwater may not be directly accessible to roots. However, the GRACE/FO signal integrates water storage across diverse vegetation types, many of which can access groundwater indirectly. For example, deeper-rooted species can redistribute water upward through hydraulic redistribution, making it available to neighboring shallow-rooted plants (e.g., Espeleta et al., 2004; Orellana et al., 2012). While this introduces some uncertainty in our $S_r$ estimates, it also reflects real-world water-sharing processes at ecosystem scales. Satellite observations further confirm

widespread plant-groundwater interactions at large spatial scales (Koirala et al., 2017), even in dryland regions dominated by grasslands (Rohde et al., 2024; T. Wang et al., 2023).

Including groundwater in our definition of $S_r$ broadens the traditional concept of the root zone, which is typically limited to the shallow unsaturated soil layer, to recognize its dynamic and functional nature (Gao et al., 2024). Many plants access deep groundwater and even rock moisture during periods of high transpiration demand or prolonged droughts, effectively expanding the functional root zone (e.g., Fan et al., 2017; McCormick et al., 2021). While this broader definition may differ from conventional understandings of the "root zone," it reflects a realistic perspective on how vegetation interacts with water resources at larger spatial scales. This expanded definition is also consistent with emerging research that incorporates groundwater as part of the water storage accessible to plants (e.g., Fan et al., 2017; McCormick et al., 2021; Miguez-Macho & Fan, 2021; Singh et al., 2020; Stocker et al., 2023).

In the revised manuscript, we refined our Introduction text in lines 35-44 to better highlight the importance of including groundwater support for ET into $S_r$. We also added a new discussion section (Section 4.3 from lines 276 - 302) to clarify how groundwater is incorporated into our $S_r^{GRACE/FO}$, and the rationale for our expanded definition of root zone storage capacity.

- Reviewer: It is highly concerning that no model attains positive NSE values for 40% of the global *vegetated* domain (Lines 216-217). This area includes most of the subtropics and Southern Hemisphere! If the model is so poor that for nearly half of the domain of interest a time-invariant constant would be a better estimator, what does it say about the skill of the model in the other half of the domain? And what does it mean for the Sr estimates in nearly half of the domain of interest where NSE is negative for all three model simulations?
  Response: In our revision, we used the GLEAM ET dataset for model validation to ensure our validation is independent of GRACE/FO and free from circularity. At least one USGS model achieved positive NSE values for about 90% of the global vegetated land (new Figs. 5 and 6), suggesting the USGS is effective in simulating ET.

Minor comments:
1. Reviewer: The heading of section 3 should probably be "Results"
   Response: Thank you. We corrected it to "Results" in the revised manuscript.

2. Reviewer: The caption of Figure 3 does not clearly state the base for the "percentage changes". This can only be understood from the text.
   Response: We changed the caption to *"(a) and (b) are the consumption percentages of $S_r^{GRACE/FO}$ during the second and third-largest TWS drawdowns."*

3. Reviewer: Line 208: Be more specific about the "drier climates and lower-biomass regions"

Response:  We specified these regions in our revised manuscript:

"*The $S_r^{GRACE/FO}$ exceeds $S_r^{accum}$ over 70% of the study area, with a median value 77 mm (or 53%) higher than that of $S_r^{accum}$, despite exhibiting lower values* **in many regions of Africa, India, Mexico, and northeast Brazil** *(Fig. 4b).*"

**References cited in Reviewer Response**

Baldocchi, D., Ma, S., & Verfaillie, J. (2021). On the inter- and intra-annual variability of ecosystem evapotranspiration and water use efficiency of an oak savanna and annual grassland subjected to booms and busts in rainfall. *Global Change Biology, 27*(2), 359-375. https://onlinelibrary.wiley.com/doi/abs/10.1111/gcb.15414

Bhanja, S. N., Mukherjee, A., Saha, D., Velicogna, I., & Famiglietti, J. S. (2016). Validation of GRACE based groundwater storage anomaly using in-situ groundwater level measurements in India. *J. Hydrol., 543*, 729-738.

Dong, J., Lei, F., & Crow, W. T. (2022). Land transpiration-evaporation partitioning errors responsible for modeled summertime warm bias in the central United States. *Nature Communications, 13*(1), 336. https://doi.org/10.1038/s41467-021-27938-6

Espeleta, J. F., West, J. B., & Donovan, L. A. (2004). Species-specific patterns of hydraulic lift in co-occurring adult trees and grasses in a sandhill community. *Oecologia, 138*(3), 341-349. https://doi.org/10.1007/s00442-003-1460-8

Esteban, E. J. L., Castilho, C. V., Melgaço, K. L., & Costa, F. R. C. (2021). The other side of droughts: wet extremes and topography as buffers of negative drought effects in an Amazonian forest. *New Phytologist, 229*(4), 1995-2006. https://nph.onlinelibrary.wiley.com/doi/abs/10.1111/nph.17005

Fan, Y., Miguez-Macho, G., Jobbágy, E. G., Jackson, R. B., & Otero-Casal, C. (2017). Hydrologic regulation of plant rooting depth. *Proceedings of the National Academy of Sciences, 114*(40), 10572-10577. https://www.pnas.org/content/pnas/114/40/10572.full.pdf

Federer, C., Vörösmarty, C., & Fekete, B. (2003). Sensitivity of annual evaporation to soil and root properties in two models of contrasting complexity. *Journal of Hydrometeorology, 4*(6), 1276-1290.

Feng, W., Zhong, M., Lemoine, J.-M., Biancale, R., Hsu, H.-T., & Xia, J. (2013). Evaluation of groundwater depletion in North China using the Gravity Recovery and Climate Experiment (GRACE) data and ground-based measurements. *Water Resources Research, 49*(4), 2110-2118. https://agupubs.onlinelibrary.wiley.com/doi/abs/10.1002/wrcr.20192

Gao, H., Hrachowitz, M., Wang-Erlandsson, L., Fenicia, F., Xi, Q., Xia, J., et al. (2024). Root zone in the Earth system. *EGUsphere, 2024*, 1-30. https://egusphere.copernicus.org/preprints/2024/egusphere-2024-332/

Getirana, A., Kumar, S., Girotto, M., & Rodell, M. (2017). Rivers and Floodplains as Key Components of Global Terrestrial Water Storage Variability. *Geophysical Research Letters, 44*(20), 10,359-310,368. https://agupubs.onlinelibrary.wiley.com/doi/abs/10.1002/2017GL074684

Ghiggi, G., Humphrey, V., Seneviratne, S. I., & Gudmundsson, L. (2021). G-RUN ENSEMBLE: A Multi-Forcing Observation-Based Global Runoff Reanalysis. *Water Resources Research, 57*(5), e2020WR028787. https://agupubs.onlinelibrary.wiley.com/doi/abs/10.1029/2020WR028787

Gou, J., & Soja, B. (2024). Global high-resolution total water storage anomalies from self-supervised data assimilation using deep learning algorithms. *Nature Water, 2*(2), 139-150. https://doi.org/10.1038/s44221-024-00194-w

Guswa, A. J. (2008). The influence of climate on root depth: A carbon cost-benefit analysis. *Water Resources Research, 44*(2). https://agupubs.onlinelibrary.wiley.com/doi/abs/10.1029/2007WR006384

Hahm, W. J., Dralle, D. N., Rempe, D. M., Bryk, A. B., Thompson, S. E., Dawson, T. E., & Dietrich, W. E. (2019). Low Subsurface Water Storage Capacity Relative to Annual Rainfall Decouples Mediterranean Plant Productivity and Water Use From Rainfall Variability. *Geophysical Research Letters, 46*(12), 6544-6553. https://agupubs.onlinelibrary.wiley.com/doi/abs/10.1029/2019GL083294

Hahm, W. J., Rempe, D., Dralle, D., Dawson, T., & Dietrich, W. (2020). Oak transpiration drawn from the weathered bedrock vadose zone in the summer dry season. *Water Resources Research, 56*(11), e2020WR027419.

Hain, C. R., Crow, W. T., Anderson, M. C., & Yilmaz, M. T. (2015). Diagnosing Neglected Soil Moisture Source–Sink Processes via a Thermal Infrared–Based Two-Source Energy Balance Model. *Journal of Hydrometeorology, 16*(3), 1070-1086. https://journals.ametsoc.org/view/journals/hydr/16/3/jhm-d-14-0017_1.xml

Hulsman, P., Keune, J., Koppa, A., Schellekens, J., & Miralles, D. G. (2023). Incorporating Plant Access to Groundwater in Existing Global, Satellite-Based Evaporation Estimates. *Water Resources Research, 59*(8), e2022WR033731. https://agupubs.onlinelibrary.wiley.com/doi/abs/10.1029/2022WR033731

Huxman, T. E., Smith, M. D., Fay, P. A., Knapp, A. K., Shaw, M. R., Loik, M. E., et al. (2004). Convergence across biomes to a common rain-use efficiency. *Nature, 429*(6992), 651-654. https://doi.org/10.1038/nature02561

Jiménez-Rodríguez, C. D., Sulis, M., & Schymanski, S. (2022). Exploring the role of bedrock representation on plant transpiration response during dry periods at four forested sites in Europe. *Biogeosciences, 19*(14), 3395-3423.

Joiner, J., & Yoshida, Y. (2020). Satellite-based reflectances capture large fraction of variability in global gross primary production (GPP) at weekly time scales. *Agricultural and Forest Meteorology, 291*, 108092.

Joiner, J., & Yoshida, Y. (2021). Global MODIS and FLUXNET-derived Daily Gross Primary Production, V2. In: ORNL Distributed Active Archive Center.

Koirala, S., Jung, M., Reichstein, M., de Graaf, I. E. M., Camps-Valls, G., Ichii, K., et al. (2017). Global distribution of groundwater-vegetation spatial covariation. *Geophysical Research Letters, 44*(9), 4134-4142. https://agupubs.onlinelibrary.wiley.com/doi/abs/10.1002/2017GL072885

Koppa, A., Rains, D., Hulsman, P., Poyatos, R., & Miralles, D. G. (2022). A deep learning-based hybrid model of global terrestrial evaporation. *Nature Communications, 13*(1), 1912. https://doi.org/10.1038/s41467-022-29543-7

Kuzyakov, Y., & Razavi, B. S. (2019). Rhizosphere size and shape: Temporal dynamics and spatial stationarity. *Soil Biology and Biochemistry, 135*, 343-360. https://www.sciencedirect.com/science/article/pii/S0038071719301452

Li, B., Rodell, M., & Famiglietti, J. S. (2015). Groundwater variability across temporal and spatial scales in the central and northeastern U.S. *Journal of Hydrology, 525*, 769-780. https://www.sciencedirect.com/science/article/pii/S0022169415002929

Li, B., Rodell, M., Kumar, S., Beaudoing, H. K., Getirana, A., Zaitchik, B. F., et al. (2019). Global GRACE data assimilation for groundwater and drought monitoring: Advances and challenges. *Water Resources Research, 55*(9), 7564-7586.

Liu, P.-W., Famiglietti, J. S., Purdy, A. J., Adams, K. H., McEvoy, A. L., Reager, J. T., et al. (2022). Groundwater depletion in California's Central Valley accelerates during megadrought. *Nature Communications, 13*(1), 7825. https://doi.org/10.1038/s41467-022-35582-x

Maxwell, R. M., & Condon, L. E. (2016). Connections between groundwater flow and transpiration partitioning. *Science, 353*(6297), 377-380. https://www.science.org/doi/abs/10.1126/science.aaf7891

McCabe, G. J., & Markstrom, S. L. (2007). *A monthly water-balance model driven by a graphical user interface* (Vol. 1088): US Geological Survey Reston, VA, USA.

McCormick, E. L., Dralle, D. N., Hahm, W. J., Tune, A. K., Schmidt, L. M., Chadwick, K. D., & Rempe, D. M. (2021). Widespread woody plant use of water stored in bedrock. *Nature, 597*(7875), 225-229. https://doi.org/10.1038/s41586-021-03761-3

McKee, T. B., Doesken, N. J., & Kleist, J. (1993). *The relationship of drought frequency and duration to time scales.* Paper presented at the Proceedings of the 8th Conference on Applied Climatology.

Miguez-Macho, G., & Fan, Y. (2021). Spatiotemporal origin of soil water taken up by vegetation. *Nature, 598*(7882), 624-628. https://doi.org/10.1038/s41586-021-03958-6

Miralles, D. G., Bonte, O., Koppa, A., Villanueva, O. B., Tronquo, E., Zhong, F., et al. (2024). GLEAM4: global land evaporation dataset at 0.1 resolution from 1980 to near present.

Miralles, D. G., Jiménez, C., Jung, M., Michel, D., Ershadi, A., McCabe, M. F., et al. (2016). The WACMOS-ET project – Part 2: Evaluation of global terrestrial evaporation data sets. *Hydrol. Earth Syst. Sci., 20*(2), 823-842. https://hess.copernicus.org/articles/20/823/2016/

Nash, J. E., & Sutcliffe, J. V. (1970). River flow forecasting through conceptual models part I— A discussion of principles. *Journal of Hydrology, 10*(3), 282-290.

Naumburg, E., Mata-gonzalez, R., Hunter, R. G., McLendon, T., & Martin, D. W. (2005). Phreatophytic Vegetation and Groundwater Fluctuations: A Review of Current Research and Application of Ecosystem Response Modeling with an Emphasis on Great Basin Vegetation. *Environmental Management, 35*(6), 726-740. https://doi.org/10.1007/s00267-004-0194-7

Novick, K. A., Ficklin, D. L., Baldocchi, D., Davis, K. J., Ghezzehei, T. A., Konings, A. G., et al. (2022). Confronting the water potential information gap. *Nature Geoscience, 15*(3), 158-164. https://doi.org/10.1038/s41561-022-00909-2

Orellana, F., Verma, P., Loheide II, S. P., & Daly, E. (2012). Monitoring and modeling water-vegetation interactions in groundwater-dependent ecosystems. *Reviews of Geophysics, 50*(3). https://agupubs.onlinelibrary.wiley.com/doi/abs/10.1029/2011RG000383

Ponce-Campos, G. E., Moran, M. S., Huete, A., Zhang, Y., Bresloff, C., Huxman, T. E., et al. (2013). Ecosystem resilience despite large-scale altered hydroclimatic conditions. *Nature, 494*(7437), 349-352. https://doi.org/10.1038/nature11836

Reager, J. T., Thomas, B. F., & Famiglietti, J. S. (2014). River basin flood potential inferred using GRACE gravity observations at several months lead time. *Nature Geoscience, 7*(8), 588-592. https://doi.org/10.1038/ngeo2203

Rempe, D. M., & Dietrich, W. E. (2018). Direct observations of rock moisture, a hidden component of the hydrologic cycle. *Proceedings of the National Academy of Sciences, 115*(11), 2664-2669. https://www.pnas.org/doi/abs/10.1073/pnas.1800141115

Rodell, M., Famiglietti, J. S., Wiese, D. N., Reager, J. T., Beaudoing, H. K., Landerer, F. W., & Lo, M. H. (2018). Emerging trends in global freshwater availability. *Nature, 557*(7707), 651-659. https://doi.org/10.1038/s41586-018-0123-1

Rodell, M., Velicogna, I., & Famiglietti, J. S. (2009). Satellite-based estimates of groundwater depletion in India. *Nature, 460*(7258), 999-1002. https://doi.org/10.1038/nature08238

Rohde, M. M., Albano, C. M., Huggins, X., Klausmeyer, K. R., Morton, C., Sharman, A., et al. (2024). Groundwater-dependent ecosystem map exposes global dryland protection needs. *Nature, 632*(8023), 101-107. https://doi.org/10.1038/s41586-024-07702-8

Schlemmer, L., Schär, C., Lüthi, D., & Strebel, L. (2018). A Groundwater and Runoff Formulation for Weather and Climate Models. *Journal of Advances in Modeling Earth Systems, 10*(8), 1809-1832. https://agupubs.onlinelibrary.wiley.com/doi/abs/10.1029/2017MS001260

Seneviratne, S. I., Corti, T., Davin, E. L., Hirschi, M., Jaeger, E. B., Lehner, I., et al. (2010). Investigating soil moisture–climate interactions in a changing climate: A review. *Earth-Science Reviews, 99*(3), 125-161. https://www.sciencedirect.com/science/article/pii/S0012825210000139

Shamsudduha, M., & Taylor, R. G. (2020). Groundwater storage dynamics in the world's large aquifer systems from GRACE: Uncertainty and role of extreme precipitation. *Earth Syst. Dyn., 11*, 755-774.

Singh, C., Wang-Erlandsson, L., Fetzer, I., Rockström, J., & van der Ent, R. (2020). Rootzone storage capacity reveals drought coping strategies along rainforest-savanna transitions. *Environmental Research Letters, 15*(12), 124021. https://dx.doi.org/10.1088/1748-9326/abc377

Speich, M. J., Lischke, H., & Zappa, M. (2018). Testing an optimality-based model of rooting zone water storage capacity in temperate forests. *Hydrology and Earth System Sciences, 22*(7), 4097-4124.

Stocker, B. D., Tumber-Dávila, S. J., Konings, A. G., Anderson, M. C., Hain, C., & Jackson, R. B. (2023). Global patterns of water storage in the rooting zones of vegetation. *Nature Geoscience*. https://doi.org/10.1038/s41561-023-01125-2

Stoy, P. C., El-Madany, T. S., Fisher, J. B., Gentine, P., Gerken, T., Good, S. P., et al. (2019). Reviews and syntheses: Turning the challenges of partitioning ecosystem evaporation and transpiration into opportunities. *Biogeosciences, 16*(19), 3747-3775.

Thomas, B. F., Caineta, J., & Nanteza, J. (2017). Global assessment of groundwater sustainability based on storage anomalies. *Geophys. Res. Lett., 44*, 11 445-411 455.

Thompson, S. E., Harman, C. J., Konings, A. G., Sivapalan, M., Neal, A., & Troch, P. A. (2011). Comparative hydrology across AmeriFlux sites: The variable roles of climate, vegetation, and groundwater. *Water Resources Research, 47*(10). https://agupubs.onlinelibrary.wiley.com/doi/abs/10.1029/2010WR009797

Vereecken, H., Amelung, W., Bauke, S. L., Bogena, H., Brüggemann, N., Montzka, C., et al. (2022). Soil hydrology in the Earth system. *Nature Reviews Earth & Environment, 3*(9), 573-587. https://doi.org/10.1038/s43017-022-00324-6

Wang, S., Li, J., & Russell, H. A. (2023). Methods for estimating surface water storage changes and their evaluations. *Journal of Hydrometeorology, 24*(3), 445-461.

Wang, S., Li, J., & Russell, H. A. J. (2023). Methods for Estimating Surface Water Storage Changes and Their Evaluations. *Journal of Hydrometeorology, 24*(3), 445-461. https://journals.ametsoc.org/view/journals/hydr/24/3/JHM-D-22-0098.1.xml

Wang, T., Wu, Z., Wang, P., Wu, T., Zhang, Y., Yin, J., et al. (2023). Plant-groundwater interactions in drylands: A review of current research and future perspectives. *Agricultural and Forest Meteorology, 341*, 109636. https://www.sciencedirect.com/science/article/pii/S0168192323003271

Wang-Erlandsson, L., Bastiaanssen, W. G. M., Gao, H., Jägermeyr, J., Senay, G. B., van Dijk, A. I. J. M., et al. (2016). Global root zone storage capacity from satellite-based evaporation. *Hydrol. Earth Syst. Sci., 20*(4), 1459-1481. https://hess.copernicus.org/articles/20/1459/2016/

Yang, Y., Donohue, R. J., & McVicar, T. R. (2016). Global estimation of effective plant rooting depth: Implications for hydrological modeling. *Water Resources Research, 52*(10), 8260-8276. https://agupubs.onlinelibrary.wiley.com/doi/abs/10.1002/2016WR019392

Zhao, M., A, G., Liu, Y., & Konings, A. G. (2022). Evapotranspiration frequently increases during droughts. *Nature Climate Change, 12*(11), 1024-1030. https://doi.org/10.1038/s41558-022-01505-3

---

## Author Response (AR2)

March 4, 2025

Dear Editor,

Thank you for the opportunity to submit a minor revision. In response to the remaining reviewer comments on drought return period on $S_r$, we added a new discussion paragraph (lines 276 - 285) highlighting why requiring perfectly matched return periods is not necessary, given that these $S_r$ estimates are typically presented and interpretated as measures of *storage capacity* rather than time-dependent drought indicators. The added discussion clarifies the comparability between different $S_r$ products and strengthens our key conclusions. A detailed response addressing the reviewer's remaining concerns is included in this letter. In the following pages, reviewer's comments are reproduced in their entirety in black, and our responses are noted in blue.

Additionally, we have updated the Code availability and Data availability sections (lines 377 - 380) to include DOI links to our $S_r^{GRACE/FO}$ dataset and python code.

We hope you find our revised manuscript suitable for publication.

Best regards,

Meng Zhao

On behalf of all co-authors

**Reviewer 2**: In the revised version of this work, the authors have nicely addressed most of my questions. I am glad they incorporated the GLEAM ET estimates as validation as this adds robustness to the analysis.

However, I have one lingering question: in my previous comments I said that the methods used were not readily comparable. The authors argue that they are comparable because of their shared definition of the physical processes involved. I agree that there is a shared definition of the processes, but I don't think the estimates are truly comparable because:

1) Sraccum calculates Sr based on "cumulative water deficit extremes occurring with a return period of 80 years".
2) The return period for SrRDxWHC is not clear.
3) SrGRACEFO uses the maximum multiyear drydown on record.

I think that the timescales used to analyze how extreme the drydown period is will affect the storage. For example, if we assume that the return period for SrGRACEFO is a 200-years. It is reasonable to assume that Sraccum calculated for a 200-yer return period drought would be significantly larger. The estimates of Sr must be done with "comparable" droughts.

I am not 100% sure how you can directly address this issue. Perhaps if we knew the return period of the droughts you analyze in your work and then used the Sraccum estimates for that return period?

**Response**: We are glad that our effort to make the analysis more robust and improve the overall quality of this manuscript is appreciated.

Thank you for your comment regarding the comparability of $S_r$ estimates due to differences in drought return periods. While we agree that the return period over which $S_r$ is calculated will influence the values estimated, we view this return period as part of the methodology of each $S_r$ estimate. That is, because most literature interpreting prior estimations of $S_r$ does not consider the time period over which it is calculated (interpreting it instead as an absolute maximum storage capacity), our key finding - that the GRACE-derived $S_r$ shows that storage capacity is larger than previously estimated – is not sensitive to this assumption. Nevertheless, we agree that a comparison that is as similar in time period as possible would be ideal.

Additionally, the $S_r^{accum}$ used in our comparison is statistically inflated from the raw $S_r^{accum}$, which is directly derived from Earth observations from 2003-2018 before being scaled to an 80-year return period under the assumption of a Gumbel's distribution. However, it is uncertain whether real-world drought return times follow this distribution. A more appropriate comparison would be between the unscaled $S_r^{accum}$ and $S_r^{GRACE/FO}$, as the latter was derived from 2002-2022 data, a similar period as that of the unscaled $S_r^{accum}$. This comparison would ensure that both estimates reflect droughts that occurred during a similar period, rather than relying on an imposed statistical return period.

Unfortunately, the unscaled $S_r^{accum}$ is not publicly available. Even so, it is evident that the unscaled $S_r^{accum}$ would be smaller than the scaled $S_r^{accum}$ used in this study. Given that the scaled $S_r^{accum}$ is already lower than $S_r^{GRACE/FO}$, the unscaled version would be even smaller. This further supports our conclusion that $S_r^{accum}$ underestimates root-zone storage capacity compared to $S_r^{GRACE/FO}$.

We now explicitly discuss the return period assumption in $S_r^{accum}$ and its impact on the $S_r^{accum}$ - $S_r^{GRACE/FO}$ difference in lines 276 – 285, which we have reproduced below for the reviewer's convenience:

> *The discrepancy between $S_r^{GRACE/FO}$ and $S_r^{accum}$ is further influenced by differences in how drought return periods are defined. $S_r^{accum}$ values reported in Stocker et al. (2023) are statistically scaled to represent an 80-year drought return period, rather than being directly derived from observed drought events. This extrapolation assumes a fixed probability distribution of drought occurrence, which may not fully capture real-world hydrological variability. In contrast, $S_r^{GRACE/FO}$ is based on observed multiyear TWS drawdowns from 2002-2022, directly reflecting droughts that occurred over the past two decades. A more comparable approach would require using the unscaled $S_r^{accum}$, which was derived using Earth observations of precipitation and ET from 2003-2018, without statistical adjustments. However, the unscaled $S_r^{accum}$ is not publicly available. Given that the 80-year return period scaling inflates $S_r^{accum}$ values, and $S_r^{GRACE/FO}$ is still substantially larger, it follows that the unscaled $S_r^{accum}$ would be even lower, further supporting the conclusion that the $S_r^{accum}$ approach underestimates root-zone storage capacity compared to $S_r^{GRACE/FO}$.*